# ⧗ Time Is a Feature: Exploiting Temporal Dynamics in Diffusion Language Models

**Wen Wang**[1,2]*,  **Bozhen Fang**[1]*,  **Chenchen Jing**[1,3],  **Yongliang Shen**[1],  **Yangyi Shen**[4],
**Qiuyu Wang**[2],  **Hao Ouyang**[2],  **Hao Chen**[1],  **Chunhua Shen**[1,2,3]

[1] Zhejiang University, State Key Laboratory of CAD & Computer Graphics   [2] Ant Group
[3] Zhejiang University of Technology   [4] Stanford University, USA

## Abstract

Diffusion large language models (dLLMs) generate text through iterative denoising, yet current decoding strategies discard rich intermediate predictions in favor of the final output. Our work here reveals a critical phenomenon, **temporal oscillation**, where correct answers often emerge in the middle process, but are overwritten in later denoising steps. To address this issue, we introduce two complementary methods that exploit temporal consistency: 1) **Temporal Self-Consistency Voting**, a training-free, test-time decoding strategy that aggregates predictions across denoising steps to select the most consistent output; and 2) a post-training method termed **Temporal Consistency Reinforcement**, which uses Temporal Semantic Entropy (TSE), a measure of semantic stability across intermediate predictions, as a reward signal to encourage stable generations. Empirical results across multiple benchmarks demonstrate the effectiveness of our approach. Using the negative TSE reward alone, we observe a remarkable average improvement of **24.7%** on the Countdown dataset over an existing dLLM. Combined with the accuracy reward, we achieve absolute gains of **2.0%** on GSM8K, **4.3%** on MATH500, **6.6%** on SVAMP, and **25.3%** on Countdown, respectively. Our findings underscore the untapped potential of temporal dynamics in dLLMs and offer two simple yet effective tools to harness them.

Project webpage: <https://aim-uofa.github.io/dLLM-MidTruth>

## 1 Introduction

Diffusion large language models (dLLMs) (Nie et al., 2025; Zhu et al., 2025; Ye et al., 2025) have recently emerged as a promising alternative to the auto-regressive (AR) large language models, garnering significant attention for their competitive performance and potential for faster inference. In contrast to AR models, which generate text in a strictly sequential manner by predicting one token at a time, dLLMs operate through iterative cycles of denoising and remasking, predicting all masked tokens in parallel at each step. A small subset of the predicted tokens, typically those with high confidence (Nie et al., 2025), are retained, while the remaining tokens are remasked and refined in subsequent steps. *Despite their drastic architectural differences, current dLLMs typically adopt a decoding strategy that mirrors AR models: solely relying on the sequence predicted in the final denoising step as the final answer, and discarding all the intermediate predictions.*

In this work, we challenge this convention by uncovering a critical phenomenon that we term **temporal oscillation**: correct answers often appear during intermediate denoising steps but are overwritten in later iterations. This discrepancy between the final output and intermediate correctness suggests that dLLMs possess rich temporal dynamics that are largely under-utilized.

As depicted in Fig. 1, we analyze two key metrics across four widely used benchmark datasets using two representative models: LLaDA-8B-Instruct (Nie et al., 2025) and LLaDA-1.5 (Zhu et al., 2025). The first metric, final pass rate, measures the accuracy of the final output, while the second, ever-pass rate, captures whether a correct answer appears at any point during the decoding process. In Fig. 1a,

---

*WW and BF contributed equally. CS is the corresponding author.

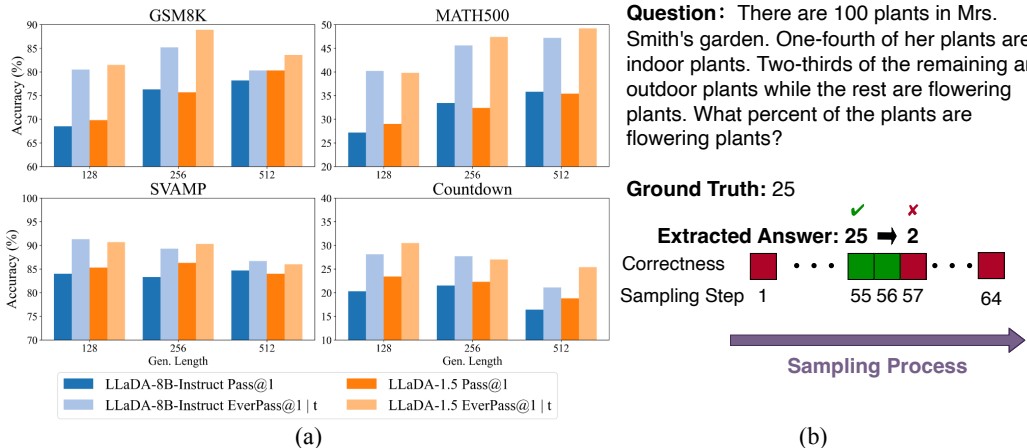

Figure 1: **Illustration of temporal oscillation during sampling.** (a) Across four datasets, a significant gap is observed between the **final answer's pass rate** (denoted as Pass @1) and the **ever-pass rate at any intermediate step** (denoted as EverPass @1 | $t$). This gap reveals the phenomenon we refer to as temporal oscillation, where correct intermediate answers are sometimes overwritten as the generation proceeds. (b) Example of temporal oscillation: For a given math problem, the model initially gives the correct answer, 25, at an intermediate step (*e.g.*, step 55), aligning with the ground truth. However, by the final step, this correct answer is replaced with an incorrect one: 2.

a consistent and significant discrepancy exists between these metrics. This gap reveals a critical phenomenon: models often generate correct answers during intermediate steps but subsequently overwrite them with incorrect ones. Fig. 1b illustrates this concretely—in a math problem, the model produces the correct answer "25" at sampling step 55, only to replace it with an incorrect "2" by the final step 64. More examples on temporal oscillation are presented in Appendix E.6.

To better understand this behavior, we analyze dLLMs from the entropy perspective and introduce a new metric: Temporal Semantic Entropy (TSE), which captures the distribution of semantic variation across intermediate outputs during decoding. Specifically, we collect the sequence of intermediate answers across denoising steps and group them into clusters based on semantic equivalence. TSE then quantifies the degree of uncertainty in the semantic content of these answers. A higher TSE indicates greater semantic fluctuation throughout the trajectory, *i.e.*, the model changes its answer frequently, while a lower TSE suggests convergence toward a stable meaning.

To harness the latent signals embedded in dLLMs decoding, we treat temporal oscillation as an informative feature and develop two complementary methods that exploit the temporal dynamics:

- **Temporal Self-Consistency Voting**: A training-free, test-time decoding strategy that aggregates predictions across multiple denoising steps and selects the most temporally consistent output. This simple yet effective method improves accuracy while introducing only negligible computational overhead, making it practical for real-world deployment.

- **Temporal Consistency Reinforcement**: A post-training method based on reinforcement learning that uses negative TSE as a reward signal to encourage stable and consistent generations explicitly. Notably, leveraging negative TSE as the reward enables reliable performance improvements without requiring ground-truth labels for reward computation. Furthermore, when ground-truth labels are available, combining them with TSE-based rewards can provide richer and more complementary supervision, ultimately leading to even greater and more robust improvements in generation quality.

Experiments across multiple datasets validate the effectiveness of both our decoding-time strategy and our RL-based post-training method. Specifically, Temporal Self-Consistency Voting brings an average improvement of **1.5%** over the LLaDA-8B-Instruct baseline with negligible overhead. In terms of Temporal Consistency Reinforcement, fine-tuning using the negative TSE reward alone, we observe a substantial average improvement of **24.7%** on the Countdown dataset. When combined with the accuracy reward derived from ground truth, our approach yields notable improvements

across diverse datasets: **2.0%** on GSM8K, **4.3%** on MATH500, **6.6%** on SVAMP, and an impressive **25.3%** on Countdown, respectively. By quantifying and leveraging temporal consistency, we offer a new perspective on dLLM decoding and introduce practical tools to unlock their potential. We hope that this study inspires further research into the temporal characteristics of diffusion decoding.

## 2 RELATED WORK

**Diffusion language models.** Building on the success of diffusion in image and video generation (Song et al., 2020; Ho et al., 2020; 2022), diffusion methods have been extended to text. Early continuous approaches (Han et al., 2022; Li et al., 2022) operate in continuous space, while others map text to the probability simplex (Avdeyev et al., 2023; Stark et al., 2024). More recent work applies flow matching on the simplex to learn categorical distributions (Davis et al., 2024; Cheng et al., 2024), but remains limited to simpler sequences. Discrete diffusion models, pioneered by D3PM (Austin et al., 2021), advanced through masked token frameworks (Shi et al., 2024; Sahoo et al., 2024; Nie et al., 2024) and scaling efforts. Lightweight variants like Plaid (Gulrajani & Hashimoto, 2023) and SEDD (Lou et al., 2023) rival GPT-2 (Radford et al., 2019), yet lag autoregressive models in scalability. To bridge this gap, BD3-LMs (Arriola et al., 2025) and Eso-LMs (Sahoo et al., 2025) interpolate between autoregressive and diffusion paradigms, enabling parallel sampling with competitive performance. Recent efforts scale up dLLMs: Dream (Ye et al., 2025) converts pretrained autoregressive models into diffusion models, while LLaDA (Nie et al., 2025) trains strong models from scratch. Another line of work studies sampling in dLLMs. For instance, Kim et al. (2025) show token ordering affects performance and propose adaptive inference, while ReMDM (Wang et al., 2025) uses inference-time remasking to boost generation. Rather than modifying local sampling heuristics, we study dLLMs through the underexplored lens of temporal stability across the entire denoising trajectory.

Several concurrent work also investigates temporal behaviors in diffusion language models. Li et al. (2025a) shows that dLLMs often generate correct intermediate answers that are later overwritten, and propose stopping generation early when such signals appear. He et al. (2025) further demonstrates that temporal cues along the denoising trajectory can be directly exploited to improve reasoning. Xie et al. (2025) introduces a step-aware RL method that aligns denoising steps with hierarchical reasoning to avoid unstructured refinement. Differently, our work takes a complementary direction by developing both a lightweight test-time voting method and an unsupervised reinforcement signal grounded in trajectory-level temporal consistency.

**Test-time Strategy.** Test-time strategies (Wei et al., 2022; Madaan et al., 2023; Snell et al., 2024; Yao et al., 2023; Liu et al., 2025a) are widely used to improve LLM accuracy, consistency, and reliability. A simple yet effective method is Self-Consistency (Wang et al., 2022), which selects the most consistent answer from multiple outputs via majority voting. Building on this idea, we propose a temporal self-consistency strategy tailored for dLLMs, which adds negligible inference overhead and integrates seamlessly into existing frameworks. Another important technique is semantic entropy (Farquhar et al., 2024; Kuhn et al., 2023), an uncertainty metric that clusters semantically equivalent outputs before computing entropy. While previously applied to uncertainty estimation and hallucination detection, we extend it to dLLMs by introducing Temporal Semantic Entropy, capturing stability and confidence throughout the denoising process.

**Post-Training using Reinforcement Learning.** Group Relative Policy Optimization (Shao et al., 2024; Guo et al., 2025) (GRPO), a variant of Proximal Policy Optimization (Schulman et al., 2017), computes advantages directly from group rewards, removing the need for a separately trained value function. GRPO has shown strong performance in reasoning tasks like mathematics and code generation, and promising results across broader modalities (Huang et al., 2025; Shen et al., 2025; Qi et al., 2025; Zhong et al., 2025; Li et al., 2025b; Damani et al., 2025). Building on this, refinements such as DAPO (Yu et al., 2025) introduce dynamic sampling to balance training batches, while entropy-based methods (Zhang et al., 2025; Prabhudesai et al., 2025; Cui et al., 2025; Agarwal et al., 2025) further enhance RL. For example, EMPO (Zhang et al., 2025) derives rewards from semantic entropy, and Seed-GRPO (Chen et al., 2025) improves advantage estimation. Recent adaptations to dLLMs include *diffu*-GRPO (Zhao et al., 2025), UniGRPO (Yang et al., 2025), and coupled-GRPO (Gong et al., 2025), which still rely on ground-truth rewards. By contrast, our method is fully unsupervised, enhancing temporal consistency without ground-truth supervision.

## 3 EXPLORATIONS ON DLLMS

### 3.1 PRELIMINARIES ON DLLMS

DLLMs formulate text generation as a process of iteratively denoising text sequences across different time steps. Let $\mathbf{x}_0 \sim p_{\text{data}}(\mathbf{x}_0)$ denote the original clean input sequence. For each diffusion step $t \in [0, T]$, let $\mathbf{x}_t \in \mathcal{V}^L$ denote the corresponding noisy token-sequence tensor of length $L$. The noisy sequence $\mathbf{x}_t$ is generated via a masking-based forward corruption process, in which a subset of tokens is stochastically masked at each step. The forward noising process is defined as a Markov chain $q(\mathbf{x}_{1:T} \mid \mathbf{x}_0) = \prod_{t=1}^{T} q(\mathbf{x}_t \mid \mathbf{x}_{t-1})$, which progressively adds noise to $\mathbf{x}_0$ over time steps. This process incrementally transforms the clean sequence $\mathbf{x}_0$ into a highly noisy version $\mathbf{x}_T$ through a series of conditional transitions. In contrast, the reverse (generative) process is modeled as:

$$p_\theta(\mathbf{x}_{0:T}) = p_\theta(\mathbf{x}_T) \prod_{t=1}^{T} p_\theta(\mathbf{x}_{t-1} \mid \mathbf{x}_t) = p_\theta(\mathbf{x}_T) \prod_{t=1}^{T} \left[ \sum_{\mathbf{x}_0^t} q(\mathbf{x}_{t-1} \mid \mathbf{x}_0^t) \, p_\theta(\mathbf{x}_0^t \mid \mathbf{x}_t) \right], \tag{1}$$

where the summation $\sum_{\mathbf{x}_0^t}$ enumerates all possible decoded clean sequences $\mathbf{x}_0^t$ at step $t$, making explicit that the sampling procedure first predicts a clean intermediate sequence and then re-applies noise via $q(\mathbf{x}_{t-1} \mid \mathbf{x}_0^t)$.

This decomposition reveals that each reverse step consists of two sub-operations aligned with the marginalization in Eq. (1). First, the model predicts a clean latent sequence through $p_\theta(\mathbf{x}_0^t \mid \mathbf{x}_t)$, yielding an intermediate hypothesis $\mathbf{x}_0^t$. Second, the next state $\mathbf{x}_{t-1}$ is generated by sampling from $q(\mathbf{x}_{t-1} \mid \mathbf{x}_0^t)$, which re-applies noise to $\mathbf{x}_0^t$ following the forward corruption process. This remasking mechanism is realized in practice through strategies such as random or low-confidence re-masking, as adopted in (Nie et al., 2025).

### 3.2 TEMPORAL OSCILLATION

In the reverse process of dLLMs, predictions $p_\theta(\mathbf{x}_0^t \mid \mathbf{x}_t)$ at a single step are often inaccurate, especially under high noise when $t$ is large. Existing models perform iterative denoising according to the generative process framework, where the final output is determined by the prediction $\mathbf{x}_0^1 \sim p_\theta(\mathbf{x}_0^1 \mid \mathbf{x}_1)$ at the last denoising step, while neglecting all intermediate predictions $\{ \mathbf{x}_0^t \sim p_\theta(\mathbf{x}_0^t \mid \mathbf{x}_t) \}_{t=2}^{T}$ generated during the iterative process. In this work, we conduct an in-depth investigation into these intermediate-step results, revealing a critical phenomenon in diffusion-based text generation.

To formalize our analysis, let $e_{i,k}$ denote the Pass@1 rate (Chen et al., 2021) for the prediction generated at the $k$-th noise step on the $i$-th question in the evaluation benchmark. Building on this, we introduce the ever pass rate , denoted as EverPass@1 | $t$, which measures the proportion of questions in the dataset for which the model produces a correct answer at *any* timestep along the sampling trajectory. Formally, it is defined as:

$$\text{EverPass@1} \mid t = \mathbb{E}_i \left\{ \max_{k \in \{1,\dots,t\}} e_{i,k} \right\} \tag{2}$$

This metric captures the cumulative correctness across all sampling steps, reflecting the overall fraction of questions for which the model arrives at a correct solution at least once, even if that solution is later discarded in the final output.

**Experiment Setup.** We compare the final pass rate, *i.e.*, Pass@1 at the last step with EverPass@1 | $t$ on two representative dLLMs: LLaDA-8B-Instruct and LLaDA-1.5, evaluated across different answer lengths and four reasoning benchmarks: GSM8K (Cobbe et al., 2021), MATH500 (Lightman et al., 2023), SVAMP (Patel et al., 2021), and Countdown (Pan et al., 2025).

**Observations.** As shown in Fig. 1 and Table 1, there is a notable gap between the final pass rate and the ever pass rate . For instance, on GSM8K with length 128, LLaDA-8B-Instruct achieves 68.5% final pass rate versus 80.5% ever pass rate , a gap of 12.0%. This gap shows that many questions are correctly solved at intermediate steps but later revised to incorrect answers during refinement. It reveals an instability in iterative decoding, where correct paths can be overwritten as generation proceeds. We term this phenomenon *temporal oscillation*, with examples in Appendix E.6.

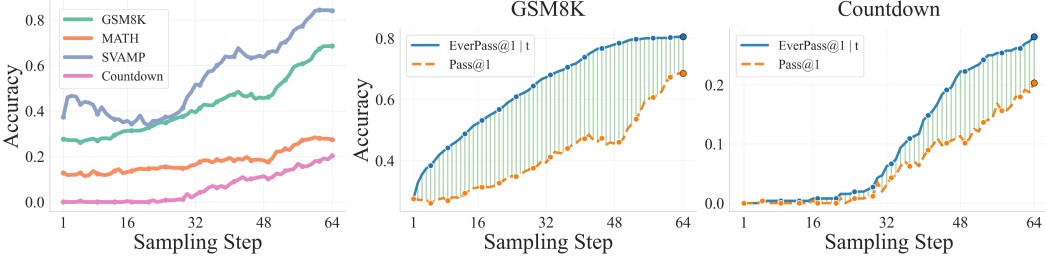

Figure 2: **Patterns of accuracy evolution over diffusion sampling steps.** Responses of length 128 are generated with 64 steps using LLaDA-8B-Instruct. **Left**: Accuracy generally rises with more steps across datasets; SVAMP starts high, while harder ones like Countdown start low but improve steadily. **Middle/Right**: We compare the final pass rate, $\mathrm{Pass}@1$, with cumulative $\mathrm{EverPass}@1 \mid t$ over steps. A clear gap persists between them, shown by the green shaded area.

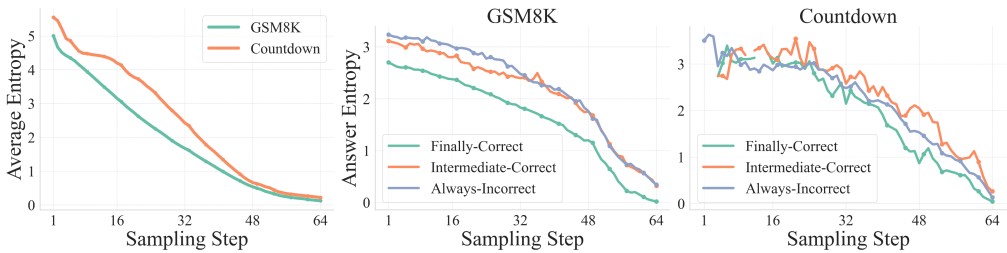

Figure 3: **Patterns of entropy evolution over diffusion sampling steps.** Responses are generated with length 128 using 64 diffusion steps from the LLaDA-8B-Instruct model. **Left:** Average token-level entropy decreases steadily during sampling. GSM8K shows lower entropy than Countdown, aligning with its higher accuracy. **Middle and Right:** Both Intermediate-Correct and Always-Incorrect questions exhibit higher overall entropy compared to Finally-Correct ones. On GSM8K, Intermediate-Correct questions display lower entropy in the early steps than Always-Incorrect, indicating initial confidence, whereas on Countdown the entropy trend is less stable.

> **Takeaway 1: Correct intermediate answers may be lost during sampling**
>
> During sampling, answers may oscillate between correct and incorrect states across diffusion steps. A notable portion of questions achieve correct answers in the intermediate steps, but ultimately yield incorrect results in the final step.

## 3.3 ANALYSES

To gain a deeper understanding of the temporal oscillation phenomenon, we conduct comprehensive analyses from multiple dimensions: accuracy, entropy, and semantic stability across decoding steps.

**Accuracy Across Sampling Steps.** As shown in Fig. 2a, accuracy generally improves with more sampling steps. Simpler datasets like SVAMP start high and remain stable, while harder ones like Countdown begin low but benefit from iterative refinement. To probe further, we compare $\mathrm{Pass}@1$ and $\mathrm{EverPass}@1 \mid t$ on GSM8K and Countdown. Early correct predictions appear sooner on GSM8K but later on Countdown, and a growing gap between the two metrics reveals that early correctness does not ensure stable reasoning. This underscores the importance of preserving correct intermediate states. Additional results for SVAMP and MATH500 are in Appendix C.1.

**Entropy Across Sampling Steps.** Temporal oscillations reflect model uncertainty. To quantify this, we analyze the average token-level entropy. As shown in Fig. 3, entropy decreases steadily and approaches zero by the final step, with GSM8K starting lower, indicating higher initial confidence. We categorize questions into three groups: Finally-Correct, Always-Incorrect, and Intermediate-Correct, where Intermediate-Correct means at least one intermediate step is correct

but the final answer is incorrect. For GSM8K and Countdown, incorrect final answers show consistently higher entropy. On GSM8K, Intermediate-Correct questions begin with lower entropy than Always-Incorrect, suggesting initially confident but unstable predictions. Considering the semi-autoregressive sampling strategy (Nie et al., 2025), we also measure the average entropy of the currently generated block, detailed in Appendix C.3.

**Temporal Semantic Entropy.** Token-level entropy reflects local uncertainty, but we also need a measure of semantic consistency across the decoding trajectory. We therefore introduce *Temporal Semantic Entropy* (TSE), which captures the semantic variations of answers during sampling. During decoding, we obtain a sequence of $T$ intermediate answers, denoted by $\{\mathbf{x}_0^t\}_{t=1}^T$. We cluster them by semantic meaning into $\mathcal{C} = \{C_1, \ldots, C_K\}$, where each cluster $C_k$ groups answers with equivalent semantics. We define the probability mass of a semantic cluster as $P(C_k) = \sum_{\mathbf{x}_0^t} p(\mathbf{x}_0^t \in C_k) = |C_k|/T$, where $|C_k|$ counts how many intermediate answers fall into cluster $C_k$. Based on this, the TSE of a sampling trajectory is defined as:

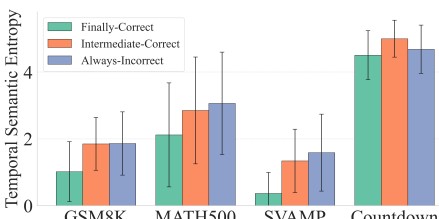

Figure 4: **Temporal semantic entropy across four benchmarks.** Error bars denote the mean ± standard deviation of TSE, clipped at 0 to respect its non-negative range. This metric measures the uncertainty in the semantic content of answers across decoding steps. Statistically, correctly answered questions exhibit lower entropy.

$$\text{TSE}(\{\mathbf{x}_0^t\}_{t=1}^T) = -\sum_{k=1}^K P(C_k) \log P(C_k), \tag{3}$$

which quantifies the uncertainty in semantic content across steps: higher TSE indicates more semantic variation, while lower TSE implies convergence to a consistent meaning.

As shown in Fig. 4, TSE offers insight into model behavior during generation. In datasets like Countdown and MATH, where performance is weaker, we observe higher entropy than in GSM8K and SVAMP, reflecting greater semantic instability. Moreover, questions ultimately answered correctly generally exhibit lower entropy than incorrect ones, including both "finally correct" and "intermediate correct" cases, indicating that stable, semantically consistent trajectories align with better performance. Thus, high TSE may signal model uncertainty and highlight samples for further improvement. More results on TSE are provided in Appendix C.2.

> **Takeaway 2: Correct answers statistically exhibit lower temporal semantic entropy**
>
> Temporal semantic entropy, computed over intermediate predictions during the decoding trajectory, reflects the semantic stability of the model's outputs. Statistically, correctly answered questions tend to have lower entropy, indicating greater consistency and confidence throughout the generation process.

## 4 METHOD

### 4.1 TEMPORAL SELF-CONSISTENCY VOTING

We propose a temporal self-consistency decoding strategy for dLLMs, which leverages intermediate predictions to improve final outputs. As discussed in Sec. 3.3, while the last timestep usually yields the best result, the correct answer may also appear earlier, so relying only on the final prediction risks discarding better outputs. To address this, we aggregate predictions across timesteps using a weighted voting mechanism. Formally, given a diffusion sampling trajectory $\{\mathbf{x}_0^t\}_{t=1}^T$, our method selects the final answer $a^*$ according to a weighted vote over all timesteps:

$$a^* = \arg\max_a \sum_{t=1}^T f(t) \cdot \mathbb{1}\left(\text{meaning}\left(\mathbf{x}_0^t\right) = a\right). \tag{4}$$

Here, $\mathbb{1}(\cdot)$ indicates whether $\mathbf{x}_0^t$ decodes to $a$, and $f(t)$ is a timestep weighting function. Since accuracy generally increases with later steps, we design $f(t)$ as a monotonically decreasing function of diffusion step. We experiment with constant, linear, and exponential weighting, as in Sec. 5.2.

**Discussion.** Our method is conceptually related to self-consistency decoding (Wang et al., 2022), which improves reasoning in autoregressive LLMs by sampling diverse reasoning paths and selecting the most consistent answer via majority voting. However, self-consistency requires multiple full-length forward passes with high cost. In contrast, our approach requires only a single sampling trajectory. By exploiting the temporal nature of diffusion inference indicated by Eq. (1), we obtain a series of intermediate predictions without additional model evaluations. This makes our method both efficient and effective for boosting accuracy through temporal aggregation.

## 4.2 TEMPORAL CONSISTENCY REINFORCEMENT

Motivated by our observation in Sec. 3.3 that correct answers generally exhibit lower Temporal Semantic Entropy (TSE) than incorrect ones, reflecting stronger semantic consistency over time, we propose a post-training approach designed to encourage temporal consistency in model outputs. Specifically, we adopt Group Relative Policy Optimization (GRPO) (Shao et al., 2024; Guo et al., 2025) as our reinforcement learning framework and use TSE as a self-supervised reward signal.

**Negative TSE as the Reward.** Following GRPO, for each question $q$ sampled from the dataset $\mathcal{D}$, we draw a group of $G$ responses $\{\mathbf{o}_1, \mathbf{o}_2, \ldots, \mathbf{o}_G\}$ from the old policy $\pi_{\theta_{old}}$. Each response $\mathbf{o}_i$ receives a scalar reward $r_i = -\text{TSE}(\mathbf{o}_i)$, where $\text{TSE}(\mathbf{o}_i)$ is computed using Eq. (3) from Sec. 3.3. This reward encourages the model to produce responses whose intermediate predictions remain semantically consistent throughout the decoding process. Based on this, we define the unnormalized advantage (Liu et al., 2025b) for all tokens $k = 1, \ldots, |\mathbf{o}_i|$ as $A_i^k(\pi) = r_i(\pi) - \text{mean}\left(\{r_j(\pi)\}_{j=1}^G\right)$. The training objective follows the standard GRPO formulation with our TSE-based reward. We optimize the policy model by maximizing the following objective:

$$\mathcal{J}_{\text{GRPO}}(\theta) = \mathbb{E}_{q \sim \mathcal{D} \atop \mathbf{o}_1, \ldots, \mathbf{o}_G \sim \pi_{\theta_{old}}(\cdot|q)} \left[ \left( \frac{1}{G} \sum_{i=1}^G \frac{1}{|\mathbf{o}_i|} \sum_{k=1}^{|\mathbf{o}_i|} \min\left(\rho_i^k A_i^k, \text{clip}\left(\rho_i^k, 1-\varepsilon, 1+\varepsilon\right) A_i^k\right) \right) - \beta D_{\text{KL}}\left[\pi_\theta(\cdot|q)\|\pi_{\text{ref}}(\cdot|q)\right] \right], \quad (5)$$

where $\pi_{\text{ref}}$ is the reference policy, $\rho_i^k = \frac{\pi_\theta(\mathbf{o}_i^k|q)}{\pi_{\theta_{old}}(\mathbf{o}_i^k|q)}$ is the importance sampling ratio, $\varepsilon$ is the clipping threshold, and $\beta$ controls the strength of the KL penalty. To compute the token-level probabilities used in $\rho_i^k$, we follow the *diffu*-GRPO method (Zhao et al., 2025), which estimates these probabilities by averaging outputs from multiple randomly masked versions of the prompt.

**Combining TSE with Accuracy Reward.** When ground-truth answers are available during training, we combine TSE with an accuracy reward to further enhance performance. Specifically, We design a composite reward function that integrates correctness and temporal consistency.

The accuracy reward follows a binary scheme: it assigns 1 when the model's prediction is correct ($\mathbf{o}_i = \mathbf{o}^*$), and 0 otherwise. To measure temporal consistency, we derive a normalized confidence score from TSE: $c(\mathbf{o}_i) = \frac{\mathcal{H}_{\max} - \text{TSE}(\mathbf{o}_i)}{\mathcal{H}_{\max}}$, $\mathcal{H}_{\max} = \log T$. Here, $T$ denotes the total sampling steps. This normalization yields $c \in [0, 1]$, with larger values indicating stronger consistency.

We further transform $c(\mathbf{o}_i)$ using the spherical scoring rule (Gneiting & Raftery, 2007), which has shown strong empirical performance across tasks. The final reward is defined as:

$$r_i = \mathbb{1}_{\mathbf{o}_i = \mathbf{o}^*} + \frac{c(\mathbf{o}_i)}{\sqrt{(c(\mathbf{o}_i))^2 + (1 - c(\mathbf{o}_i))^2}}. \quad (6)$$

Here, the first term enforces correctness while the second term encourages higher temporal consistency. This formulation makes the reward sensitive not only to prediction accuracy, but also to the stability of the underlying generation process. A detailed comparison of various scoring rules and their empirical performance in the Appendix E.4.

**Discussion.** Unlike prior reinforcement learning post-training methods for dLLMs, such as *diffu*-GRPO, which relies on ground-truth answers for reward computation, our approach operates without any labeled data. Instead, we harness the model's internal temporal dynamics as a self-supervised signal, employing negative TSE to assess answer quality. This enables our method to be broadly applicable, particularly in unsupervised settings, and offers a novel direction for improving dLLMs. Furthermore, we show that combining the negative TSE reward with the accuracy reward based on ground-truth answers yields notably better performance than using the accuracy reward alone.

Table 1: **Performance of temporal majority voting.** We compare three strategies: fixed, linear, and exponential weighting, on four datasets using LLaDA-8B-Instruct and LLaDA-1.5. Bold numbers mark group bests, and green values show gains over baseline. For reference, we report the oracle EverPass @1 $\mid t$ as an upper bound.

| | | GSM8K | | | MATH500 | | | SVAMP | | | Countdown | | |
|---|---|---|---|---|---|---|---|---|---|---|---|---|---|
| | Method / Seq Len | 128 | 256 | 512 | 128 | 256 | 512 | 128 | 256 | 512 | 128 | 256 | 512 |
| **LLaDA-8B-Instruct** | baseline | 68.5 | 76.3 | 78.2 | 27.4 | 33.4 | 35.8 | 84.0 | 83.3 | **84.7** | 20.3 | 21.5 | **16.4** |
| | Fixed Weighting | 68.0 | 73.4 | 78.3 | 26.6 | 30.8 | 34.2 | **87.0** | 84.3 | 84.3 | 22.7 | 18.8 | 11.3 |
| + Temporal Voting | Linear Weighting | 70.0 | 78.0 | 78.8 | 28.0 | 34.4 | 34.6 | **87.0** | 84.3 | 84.3 | 24.2 | 21.9 | 16.0 |
| | Exp. Weighting | **70.1** | **78.7** | **78.9** | **28.4** | **35.6** | **36.2** | 86.0 | **84.3** | **84.7** | **25.0** | **23.4** | **16.4** |
| | | +1.6 | +2.4 | +0.7 | +1.0 | +2.2 | +0.4 | +2.0 | +1.0 | +0.0 | +4.7 | +1.9 | +0.0 |
| | EverPass @1 $\mid t$ | 80.5 | 85.2 | 80.3 | 40.2 | 45.6 | 47.2 | 91.3 | 89.3 | 86.7 | 28.1 | 27.7 | 21.1 |
| **LLaDA-1.5** | baseline | 69.8 | 79.4 | **81.1** | 29.0 | 32.4 | 35.4 | 85.3 | 86.3 | 83.3 | 21.5 | 21.1 | 20.7 |
| | Fixed Weighting | 68.8 | 75.7 | 80.3 | 27.3 | 30.8 | 34.6 | **87.3** | 85.3 | 84.0 | 23.4 | 22.3 | 18.8 |
| + Temporal Voting | Linear Weighting | **71.0** | **79.8** | 81.0 | **29.2** | 32.8 | 35.8 | 86.0 | 87.0 | 84.0 | 24.2 | 23.4 | 19.1 |
| | Exp. Weighting | 70.7 | **79.8** | **81.1** | 29.0 | **33.2** | **36.2** | 85.7 | **87.7** | **84.3** | **26.2** | **25.0** | **21.1** |
| | | +0.9 | +0.4 | +0.0 | +0.0 | +0.8 | +0.8 | +0.4 | +1.4 | +1.0 | +4.7 | +3.9 | +0.4 |
| | EverPass @1 $\mid t$ | 81.5 | 88.9 | 83.6 | 39.8 | 47.4 | 49.2 | 90.7 | 90.3 | 86.0 | 30.5 | 27.0 | 25.4 |

# 5 EXPERIMENTS

## 5.1 IMPLEMENTATION DETAILS

**Experimental Setup.** We use LLaDA-8B-Instruct (Nie et al., 2025) and LLaDA-1.5 (Zhu et al., 2025), evaluating on four math benchmarks: GSM8K (Cobbe et al., 2021), MATH500 (Hendrycks et al., 2021), SVAMP (Patel et al., 2021), and Countdown (Pan et al., 2025). Following d1 (Zhao et al., 2025), we report performance under different output lengths. Temporal self-consistency voting applies exponential weights to sampling steps. All of our experiments adopt semi-autoregressive decoding with low-confidence remasking, using a block size of 32. The diffusion step is set to half of the target output length. For post-training, LLaDA-8B-Instruct undergoes supervised fine-tuning (SFT) on s1K (Muennighoff et al., 2025) for 20 epochs with 4,096 token sequences, followed by reinforcement fine-tuning (RFT). For LLaDA-1.5, we omit SFT because it often results in performance degradation, likely due to the model having already undergone sophisticated post-training. All training uses 8 H800 GPUs. Further details are in Appendix B.

**Answer Extraction for Voting and Semantic Clustering.** At every diffusion step, LLaDA first generates a fully decoded token sequence *before* applying the remasking operation. Although some positions may be remasked and re-predicted in later steps, this pre-remask sequence is always a complete, unmasked prediction. We use these fully decoded sequences as the intermediate outputs for all temporal analyses. This design guarantees that each intermediate prediction provides a syntactically complete candidate answer. Answer extraction itself is performed by parsing the `<answer>...</answer>` block. If a pre-remask sequence does not contain a valid answer span at a particular step, that step is excluded from temporal self-consistency voting or semantic clustering. Only steps containing valid, fully parsed answers contribute to our temporal statistics.

## 5.2 TEMPORAL SELF-CONSISTENCY VOTING

**Voting Strategies.** We apply weighted voting across denoising steps using three schemes: fixed, linear, and exponential. Each uses a weighting function $f(t)$, where $t$ is the current diffusion step. The fixed scheme assigns equal weight to all steps with $f(t) = 1$. The linear weighting takes the form $f(t) = 1 - t/T$, and exponential uses $f(t) = \exp(\alpha(1 - t/T))$ with $\alpha = 5$. Both linear and exponential schemes prioritize early diffusion time steps, *i.e.*, latter sampling steps.

**Ablations on Voting Strategies.** As shown in Table 1, linear and exponential weighting improve inference performance, with exponential yielding the largest gains, *e.g.*, LLaDA-8B-Instruct improves by 1.6%, 1.2%, 1.0%, and 2.2% on GSM8K, MATH500, SVAMP, and Countdown, respectively. Fixed weighting performs slightly worse, likely because equal weights amplify inaccurate early predictions. We therefore adopt exponential weighting by default. We futher ablate

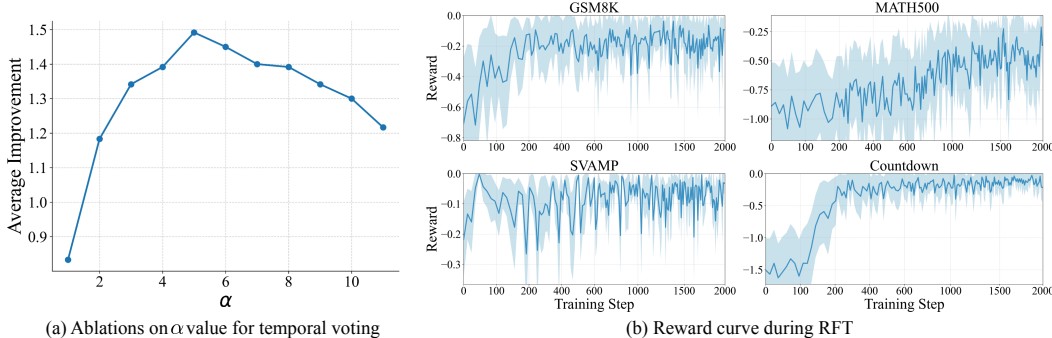

(a) Ablations on $\alpha$ value for temporal voting

(b) Reward curve during RFT

Figure 5: (a) Ablations on $\alpha$ value selection in temporal voting with exponential weighting. (b) Negative temporal semantic entropy reward curve during reinforcement fine-tuning.

Table 2: **Performance of reinforcement fine-tuning.** Unlike d1 (Zhao et al., 2025), which requires ground-truth answers, our method uses negative Temporal Semantic Entropy (TSE) as the reward without labels. Combining TSE with accuracy-based rewards yields further gains across benchmarks. Green numbers denote improvements over baseline.

| | | GSM8K | | | MATH500 | | | SVAMP | | | Countdown | | |
|---|---|---|---|---|---|---|---|---|---|---|---|---|---|
| | Method / Seq Len | 128 | 256 | 512 | 128 | 256 | 512 | 128 | 256 | 512 | 128 | 256 | 512 |
| **LLaDA** | baseline | 70.2 | 78.7 | 80.1 | 23.8 | 34.4 | 36.8 | 81.7 | 83.3 | 82.7 | 21.5 | 19.9 | 21.5 |
| | accuracy reward (d1) | 71.7 | 78.3 | 82.3 | 31.0 | **36.0** | 40.4 | **85.7** | 88.0 | 88.7 | 34.8 | 35.5 | 37.9 |
| + RFT | negative TSE reward (ours) | **72.2** | 78.8 | 80.2 | 30.6 | 34.6 | 38.0 | 84.3 | 89.0 | 88.7 | 38.6 | **53.5** | 44.9 |
| | | +2.0 | +0.1 | +0.1 | +6.8 | +0.2 | +1.2 | +2.6 | +5.7 | +6.0 | +17.1 | +33.6 | +23.4 |
| | combining both (ours) | 72.1 | **80.0** | **83.0** | **31.2** | 35.4 | **41.4** | 85.0 | **90.3** | **92.3** | **41.5** | 42.6 | **54.7** |
| | | +1.9 | +1.3 | +2.9 | +7.4 | +1.0 | +4.6 | +3.3 | +7.0 | +9.6 | +20.0 | +22.7 | +33.2 |
| **LLaDA-1.5** | baseline | 69.8 | 79.4 | 81.1 | 29.0 | 32.4 | 35.4 | 85.3 | 86.3 | 83.3 | 21.5 | 21.1 | 20.7 |
| | accuracy reward (d1) | 73.0 | 78.9 | 83.1 | 29.8 | **36.2** | 40.2 | 84.7 | 89.3 | 88.0 | 38.7 | 29.7 | 39.1 |
| + RFT | negative TSE reward (ours) | 72.0 | **80.8** | 82.6 | **30.2** | 35.0 | 40.0 | **86.3** | 88.3 | 87.3 | **50.0** | **55.9** | 53.1 |
| | | +2.2 | +1.4 | +1.5 | +1.2 | +2.6 | +4.6 | +1.0 | +2.0 | +4.0 | +28.5 | +34.8 | +32.4 |
| | combining both (ours) | **73.2** | 80.5 | **84.0** | 29.6 | 35.4 | **41.4** | **86.3** | **90.3** | **89.0** | 44.5 | 46.9 | **63.3** |
| | | +3.4 | +1.1 | +2.9 | +0.6 | +3.0 | +6.0 | +1.0 | +4.0 | +5.7 | +23.0 | +25.8 | +42.6 |

the exponential hyperparameter $\alpha$. As shown in Fig. 5a, $\alpha$ values ranging from 1 to 11 consistently improve accuracy, peaking at $\alpha = 5$ with an average gain of 1.5%. Thus, we set $\alpha = 5$ by default.

### 5.3 TEMPORAL CONSISTENCY REINFORCEMENT

**Main Results.** Table 2 reports results of incorporating temporal consistency into RFT. We have the following observations. **(1)** Using TSE reward alone consistently improves performance across lengths and datasets. **(2)** TSE reward matches or surpasses accuracy reward despite not using ground truth, *e.g.*, on Countdown, LLaDA-8B-Instruct improves 24.7% vs. 15.1% with d1. **(3)** Combining TSE with accuracy reward further boosts results, with absolute gains of 0.9% (GSM8K), 0.2% (MATH500), 1.7% (SVAMP), and 10.2% (Countdown) over d1. Overall, our method achieves average improvements of 2.0%, 4.3%, 6.6%, and 25.3% over the SFT baseline, confirming the benefit of encouraging temporal consistency.

**Training Dynamics.** We visualize the reward curves during training using LLaDA-8B-Instruct as an example, as shown in Fig. 5b. The curves demonstrate a consistent upward trend in rewards across different datasets as training progresses, indicating effective learning and stable optimization.

**Model Attributes After RFT.** We analyze LLaDA-8B-Instruct fine-tuned with the negative TSE reward, with generation length 128, evaluating its behavior across several dimensions: TSE, the ever pass rate , and the number of effective tokens (defined as the average count of non-padding, non-EOS tokens per generation). We have the following observations. **(1)** As shown in Fig. 6a, temporal semantic entropy consistently decreases across various datasets after RFT, reflecting enhanced temporal consistency in the model's outputs—an anticipated result of reinforcement learning; **(2)**

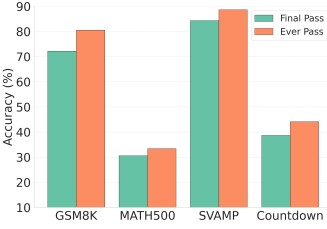 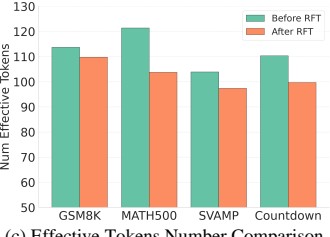

(a) Temporal Semantic Entropy Comparison  (b) Final and Ever Pass Rate After RFT  (c) Effective Tokens Number Comparison

Figure 6: **Model attributes after reinforcement finetuning (RFT).** (a) Temporal semantic entropy decreases after RFT, showing improved semantic consistency in outputs. (b) The ever pass rate remains above the final pass rate, leaving room for further gains. (c) Effective tokens per generation drop after RFT, yielding more concise outputs.

Fig. 6b demonstrates that the ever pass rate remains higher than the final pass rate after RFT, suggesting there is still potential for further improvement. **(3)** In Fig. 6c, effective tokens decline after RFT, implying more concise outputs. We suspect shorter generations may reduce temporal oscillations, though this requires further study. Further analysis of model attributes after RFT and a detailed discussion of limitations are provided in Appendix E.5 and Appendix D.

## 6 CONCLUSION

This work uncovers a critical yet overlooked aspect of diffusion large language models: their rich temporal dynamics. By identifying temporal oscillation as a recurring issue in dLLM decoding, we challenge the convention of relying solely on final-step predictions. Our proposed methods—Temporal Self-Consistency Voting and Temporal Consistency Reinforcement —demonstrate that intermediate predictions are not noise, but signal. These strategies improve accuracy and stability without requiring additional inference passes or ground-truth supervision. Through extensive experiments, we show that temporal consistency is not just a desirable property—it is a powerful lever for performance. We hope that this study inspires future research to *treat intermediate denoising time steps not as a nuisance, but as a feature in diffusion-based text generation*.

## ACKNOWLEDGMENT

This work is partially supported by the National Key R&D Program of China (No. 2022ZD01601-60101), Ningbo Science and Technology Bureau (No. 2024Z291) and the National Natural Science Foundation of China (No. 62206244). The authors would also like to thank Muzhi Zhu, Canyu Zhao, and Linhao Zhong at Zhejiang University for their valuable discussions and insightful feedback.

## ETHICS STATEMENT

This work does not involve human subjects, personal data, or sensitive information. All datasets used in our experiments (GSM8K, MATH500, SVAMP, and Countdown) are publicly available benchmark datasets designed for evaluating mathematical reasoning in large language models. We strictly adhered to ethical research practices and did not conduct any data collection that could raise privacy, security, or fairness concerns. Our methods—Temporal Self-Consistency Voting and Temporal Consistency Reinforcement—focus on improving the robustness and accuracy of diffusion language models, without introducing risks of harmful applications. To the best of our knowledge, this research complies with the ICLR Code of Ethics and poses no foreseeable ethical concerns.

## REPRODUCIBILITY STATEMENT

We have made extensive efforts to ensure the reproducibility of our work. Detailed dataset descriptions are provided in Appendix B.1, and training configurations and hyperparameters are

reported in Appendix B.2. The sampling and evaluation procedures are outlined in Appendix B.3. To further promote transparency, we provide mathematical formulations and thorough descriptions of our proposed algorithms—Temporal Self-Consistency Voting and Temporal Consistency Reinforcement—directly in the main paper. Upon acceptance, we will release our models, together with training and inference code, to facilitate replication and further research.

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

## LLM Usage

In this section, we clarify the role of large language models (LLMs) in preparing this work. The model was used exclusively for language polishing, such as refining grammar, style, and readability, without contributing to the research design, analysis, or conclusions.

## A    Appendix Overview

This appendix provides additional implementation details, empirical analysis, and extended results to supplement the main paper. It is organized as follows:

- **Appendix B: More Implementation Details**
  Provides further implementation information, including:
    - Appendix B.1: Detailed descriptions of the datasets used
    - Appendix B.2: Training configurations and hyperparameters
    - Appendix B.3: Sampling strategies and evaluation procedures
- **Appendix C: More Analysis**
  Presents extended analyses, including:
    - Appendix C.1: Accuracy and entropy analysis on the MATH500 and SVAMP datasets
    - Appendix C.2: Temporal semantic entropy across varying generated lengths
    - Appendix C.3: Block-level token entropy analysis
    - Appendix C.4: Time analysis for temporal consistency voting
    - Appendix C.5: Investigating the Causes of Time Oscillation
- **Appendix D: Limitations**
  Discuss limitations and analyze failure cases, including:
    - Appendix D.1: Discuss potential limitations of our method
    - Appendix D.2: Analysis of failure cases on the Sudoku dataset
- **Appendix E: More Experimental Results**
  Includes additional experimental findings, such as:
    - Appendix E.2: Training a unified model across multiple tasks
    - Appendix E.3: Possible alternatives to temporal semantic entropy
    - Appendix E.4: Ablation studies on different scoring rules for combining TSE with accuracy reward
    - Appendix E.5: Performance of temporal self-consistency voting on reinforcement fine-tuned models
    - Appendix E.6: Detailed examples illustrating temporal oscillation

## B    More Implementation Details

### B.1    Datasets

We provided detailed descriptions of the datasets as follows:

- GSM8K (Cobbe et al., 2021) comprises 8.5K linguistically diverse grade school math word problems (7.5K training, 1K test), solvable by bright middle school students via 2–8 steps of basic arithmetic, suited for multi-step mathematical reasoning.
- MATH500 (Lightman et al., 2023) is a curated subset of 500 problems selected from the broader MATH dataset (Hendrycks et al., 2021), featuring high-school-level competition math problems.
- SVAMP (Patel et al., 2021) serves as a benchmark for elementary-level Math Word Problems (MWPs), where each MWP is a short natural language narrative describing a scenario and asking questions about unknown quantities.

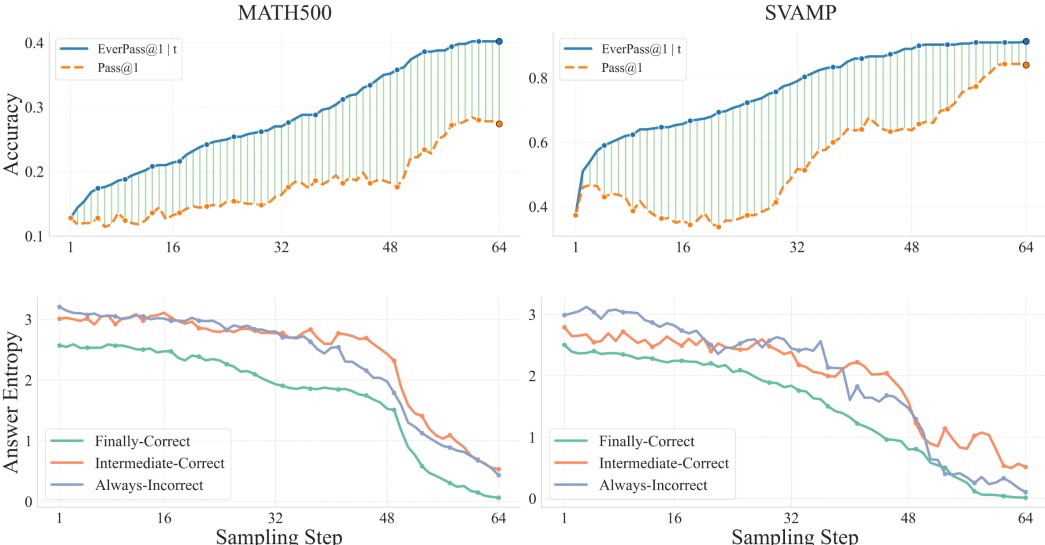

Figure S1: **Top Row:** Pass @1 and Pass @1 | $t$ for MATH500 and SVAMP are provided as supplementary results to Fig. 2, using the same experimental settings. A noticeable gap similar to that observed in GSM8K and Countdown between Pass @1 and Pass @1 | $t$ is present across all sampling steps. **Bottom Row:** Answer Entropy for MATH500 and SVAMP are provided as supplementary results to Fig. 3, using the same experimental setups. MATH500 and SVAMP exhibit answer entropy patterns similar to those of GSM8K and Countdown.

- Countdown (Pan et al., 2025) involves a combinatorial arithmetic game with three numbers, requiring models to reach target numbers using basic arithmetic operations on a given set of numbers.

## B.2 TRAINING

During reinforcement fine-tuning, we train our model using sequences of 256 tokens, with a batch size of 6 per GPU and gradient accumulation over 2 steps. Low-Rank Adaptation (LoRA) (Hu et al., 2022) is applied with a rank of 128 and a scaling factor of 64. During reward computation, answers are parsed from generated text sequences for semantic clustering. When answer parsing fails due to an inaccurate format, we simply discard the answer for temporal semantic entropy computation. Moreover, since answers generated in the first half of the sampling steps tend to be rough and less reliable, we exclude them from consideration. Only answers from the second half of the sampling steps are used to calculate the temporal semantic entropy.

## B.3 SAMPLING AND EVALUATION

**Unified Decoding Configuration.** During sampling, we adopt the semi-autoregressive sampling approach following LLaDA (Nie et al., 2025). Specifically, the sequence is split into multiple blocks, which are generated in a left-to-right manner. For each individual block, we employ the low-confidence remasking strategy during the sampling process. We use a block size of 32, following LLaDA (Nie et al., 2025) and d1 (Zhao et al., 2025). Following the practice in d1 (Zhao et al., 2025), we evaluate the model every 100 steps, starting from step 600 to 8,000 steps, and report the best results.

**Generation Lengths.** *(1) Evaluation.* **For test-time self-consistency voting and all RL evaluations, we evaluate three output lengths: 128, 256, and 512 tokens, corresponding to 64, 128, and 256 diffusion steps, respectively (again following d1 (Zhao et al., 2025)).** *(2) Analysis experiments.* **In the exploratory study in Sec. 3.2 / Fig. 1, we analyze output lengths of 128, 256, and 512, and observe temporal oscillation under all settings. Sec. 3.3 uses the default 128-**

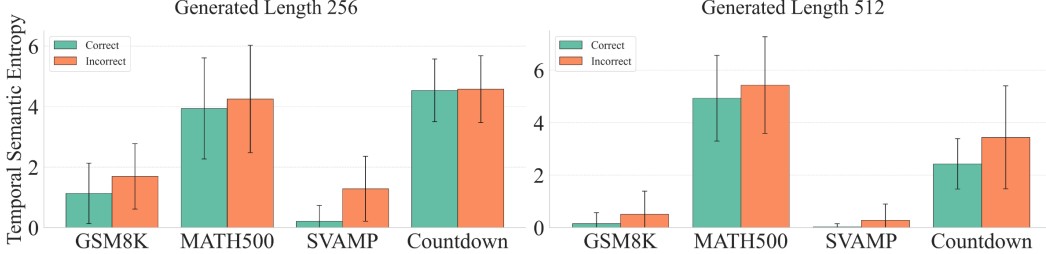

Figure S2: **Temporal semantic entropy with varying generation lengths.** Average temporal semantic entropy across datasets with generation lengths of 256 and 512, showing consistent patterns with Fig. 4. Higher entropy generally correlates with lower accuracy.

**token / 64-step configuration. Appendix Sec. C.2 / Fig. S2 further reports temporal semantic entropy under 256- and 512-length generations to demonstrate robustness.**

## C  MORE ANALYSIS

### C.1  ANALYSIS ON MATH500 AND SVAMP DATASETS

**Accuracy Analysis.** As shown in the first row in Fig. S1, MATH500 and SVAMP exhibit a similar pattern to that observed Fig. 2 in Sec. 3.3, where a noticeable gap emerges between $\text{Pass}@1$ and $\text{Pass}@1 \mid t$. In MATH500 and SVAMP, correct answers start to appear early in the sampling process(near 4% and 39% at first step, respectively), and continue to improve over subsequent iterations. Interestingly, on the SVAMP dataset, a distinct pattern emerged in the model's $\text{Pass}@1$ accuracy across different sampling steps. Between steps 3 and 20, performance declined noticeably, followed by a recovery after step 20. This distinctive fluctuation trajectory represents another manifestation of the temporal oscillation phenomenon.

**Entropy Analysis.** As shown in the second row in Fig. S1, for MATH500 and SVAMP, the answer entropy of *Finally-Correct* questions remains the lowest throughout the sampling process. *Intermediate-Correct* questions consistently exhibit lower entropy in the early sampling steps compared to Always-Incorrect ones, a pattern observed across all four datasets. However, unlike GSM8K, MATH500, and Countdown, where the final entropy of *Intermediate-Correct* and *Always-Incorrect* questions is similar, SVAMP displays relatively high entropy in the final sampling step for *Intermediate-Correct* questions.

### C.2  TEMPORAL SEMANTIC ENTROPY OVER VARYING GENERATED LENGTH

To further validate the generalizability of our findings regarding temporal semantic entropy, we extended the experiments beyond those presented in Fig. 4, which used a generation length of 128 and 64 diffusion steps. Specifically, we tested generation lengths of 256 and 512, with diffusion steps set to half the generation length.

As shown in Fig. S2, the pattern of temporal semantic entropy observed here aligns with the conclusions drawn in Sec. 3.3: questions that are eventually answered correctly tend to display consistently low temporal semantic entropy, indicating that their intermediate predictions remain stable and semantically coherent throughout the decoding process.

### C.3  ANALYSIS ON THE AVERAGE ENTROPY IN BLOCKS

In addition to the token-level entropy considered in Sec. 3.3 and the answer-level entropy considered in temporal semantic entropy, we additionally introduce a block-level entropy. This is motivated by the fact that dLLMs typically adopts a semi-autoregressive sampling strategy. In this sampling strategy, the entire generated sequence is divided into multiple fixed-length blocks, with each block allocated a specific number of sampling steps. During these steps, remasking and unmasking

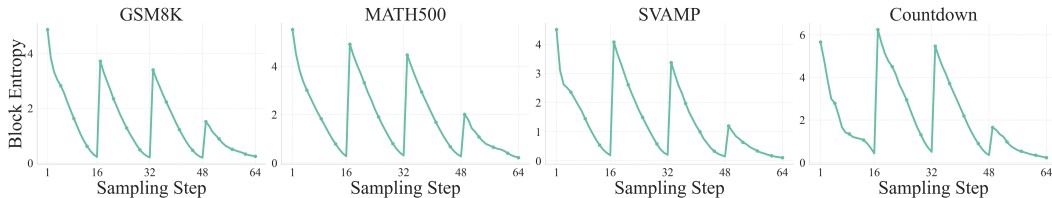

Figure S3: **Block-level entropy dynamics under semi-autoregressive sampling.** During sampling, sequences are partitioned into fixed-length blocks, processed in a left-to-right order, with remasking and unmasking operations restricted to the current block. Average token entropy within a block decreases with more sampling steps. A sharp entropy spike occurs when shifting to a new block, likely due to simultaneous decoding of multiple masked tokens increasing initial uncertainty.

Table S1: **Runtime comparison (in seconds) for temporal self-consistency voting.** For the voting setting, we report end-to-end latency from receiving the input to producing the final voted output. For the non-voting setting, we measure the time required to generate an output of the same final length. Across all datasets, the additional cost introduced by voting is minimal.

| | Method / Seq Len | GSM8K | | | MATH500 | | | SVAMP | | | Countdown | | |
|---|---|---|---|---|---|---|---|---|---|---|---|---|---|
| | | 128 | 256 | 512 | 128 | 256 | 512 | 128 | 256 | 512 | 128 | 256 | 512 |
| LLaDA-8B-Instruct | *w/o* voting (baseline) | 4.48 | 10.76 | 29.24 | 4.47 | 10.87 | 29.45 | 4.23 | 10.31 | 28.45 | 4.59 | 10.99 | 29.58 |
| | *w/* voting | 4.53 | 10.86 | 29.76 | 4.54 | 10.91 | 29.61 | 4.29 | 10.41 | 28.59 | 4.64 | 11.12 | 29.81 |

operations are performed exclusively on the current block. Starting with the first block, the process steps to the next block only after all tokens in the current block have been unmasked.

We denote the current block as containing tokens indexed from a start $s$ to an end $e - 1$ (inclusive), such that the block spans indices $i \in \{s, s + 1, \ldots, e - 1\}$. The block entropy is then calculated as:

$$H_{\text{block}} = \frac{1}{e - s} \sum_{i=s}^{e-1} H(i)$$

where $H(i)$ represents the entropy of the $i$-th generated token, and $e - s$ denotes the total number of tokens in the block.

As illustrated in Fig. S3, the average token entropy within a single block exhibits a consistent downward trend as sampling steps accumulate. This pattern is intuitive: as more tokens in the current block are decoded, they collectively form a richer contextual foundation, thereby mitigating the model's uncertainty about subsequent tokens in the block. Interestingly, when decoding shifts to a new block, the entropy rises sharply. This phenomenon is likely due to the need to decode multiple masked tokens simultaneously at the start of a new block, which increases the model's uncertainty.

## C.4 TIME ANALYSIS FOR TEMPORAL SELF-CONSISTENCY VOTING

To assess the efficiency of our temporal self-consistency voting approach, we compare the per-sample runtime with and without voting. We use LLaDA-8B-Instruct with semi-autoregressive decoding (block size 32), a diffusion step set to half of the target output length, and a batch size of 1. All timing experiments are performed on four NVIDIA 4090-24GB GPUs.

For the voting setting, we measure the end-to-end runtime from receiving the input to producing the final voted answer. For the non-voting setting, we measure the time to generate an output of the same final length. The difference reflects the additional operations performed by voting—namely answer parsing, clustering, and selecting the final answer. We report the average time to generate one response of the fixed output length for each dataset.

As shown in Table S1, temporal self-consistency voting introduces essentially no extra runtime. Across all datasets, the overhead is about 1% per sample—negligible relative to the total computation time.

Table S2: **Evaluation results under different randomness levels and remasking strategies.** Black numbers denote $\text{Pass}@1$, while gray numbers denote $\text{EverPass}@1|t$. As shown, the gap between $\text{Pass}@1$ and $\text{EverPass}@1|t$ persists across all seeds, temperature settings, and remasking strategies, indicating that the phenomenon of time oscillation is not sensitive to these configuration choices.

| Setting | | | GSM8K | | | MATH500 | | | SVAMP | | | Countdown | | |
|---|---|---|---|---|---|---|---|---|---|---|---|---|---|---|
| Seed | Temp. | Remasking | 128 | 256 | 512 | 128 | 256 | 512 | 128 | 256 | 512 | 128 | 256 | 512 |
| 42 | 0.0 | Low Confidence | 67.9 | 76.5 | 79.2 | 27.4 | 33.4 | 35.8 | 83.3 | 84.0 | 84.7 | 18.8 | 20.3 | 15.6 |
| | | | 80.3 | 84.8 | 81.0 | 40.2 | 45.6 | 47.2 | 90.7 | 89.7 | 87.3 | 25.8 | 27.7 | 19.9 |
| 218 | 0.0 | Low Confidence | 68.2 | 76.5 | 79.2 | 27.2 | 33.4 | 35.8 | 83.0 | 84.0 | 84.7 | 20.3 | 20.3 | 16.0 |
| | | | 81.5 | 84.8 | 81.0 | 40.8 | 45.6 | 47.2 | 91.0 | 89.7 | 87.3 | 29.3 | 27.0 | 20.7 |
| 66 | 0.0 | Low Confidence | 69.6 | 76.3 | 78.2 | 27.2 | 33.4 | 35.8 | 83.3 | 83.3 | 84.7 | 20.3 | 22.3 | 18.8 |
| | | | 80.3 | 85.2 | 80.3 | 40.8 | 45.6 | 47.2 | 90.7 | 89.3 | 86.7 | 27.7 | 29.3 | 22.7 |
| 42 | 0.6 | Low Confidence | 67.4 | 76.6 | 78.5 | 25.6 | 33.8 | 33.8 | 84.7 | 85.0 | 84.7 | 20.3 | 22.3 | 18.8 |
| | | | 82.6 | 86.7 | 81.5 | 43.2 | 49.0 | 50.8 | 91.7 | 90.7 | 87.0 | 29.3 | 20.7 | 22.7 |
| 42 | 1.0 | Low Confidence | 66.1 | 75.2 | 77.0 | 23.8 | 30.6 | 30.8 | 85.0 | 84.3 | 85.0 | 19.5 | 17.6 | 16.8 |
| | | | 78.8 | 81.9 | 79.2 | 37.6 | 42.0 | 42.2 | 89.7 | 87.0 | 87.3 | 22.3 | 19.5 | 21.1 |
| 42 | 0.0 | Random | 57.2 | 64.5 | 62.1 | 19.8 | 24.4 | 23.4 | 73.0 | 73.3 | 72.7 | 10.9 | 8.2 | 10.5 |
| | | | 76.9 | 76.6 | 66.8 | 35.8 | 38.6 | 33.4 | 86.0 | 82.0 | 78.0 | 22.3 | 12.5 | 14.5 |
| 218 | 0.0 | Random | 59.4 | 61.6 | 62.1 | 19.8 | 24.6 | 21.2 | 75.0 | 73.3 | 72.7 | 12.1 | 9.8 | 10.5 |
| | | | 75.3 | 75.7 | 67.2 | 35.2 | 42.2 | 33.6 | 87.3 | 79.0 | 77.7 | 23.8 | 14.1 | 15.6 |

## C.5 INVESTIGATING THE CAUSES OF TIME OSCILLATION

First, we investigate the effects of randomness and temperature on time oscillation. As shown in Table S2, although both $\text{Pass}@1$ and $\text{EverPass}@1|t$ exhibit slight fluctuations across different seeds and temperature settings, the gap between them consistently persists. We further examine the impact of applying a random remasking strategy. While this strategy noticeably reduces both $\text{Pass}@1$ and $\text{EverPass}@1|t$, the gap remains. These observations suggest that time oscillation is an inherent and pervasive phenomenon originating from the model itself.

We hypothesize that time oscillation may arise from the training dynamics, where the same input signal can be associated with multiple valid outputs, which is analogous to diffusion models in the vision domain (Liu et al., 2022; Lipman et al., 2022) Specifically, consider a sequence $seq_1 =$ "$abcd$", possible that another sequence $seq_2 =$ "$abce$" also appears in the training data with the same masked pattern, and the model is likewise supervised to predict from "$<\text{MASK}>bc <\text{MASK}>$". As a result, the model receives conflicting supervision signals, causing it to oscillate between "$abcd$" and "$abce$" during sampling.

Moreover, even if no alternative sequence like $seq2$ exists, the model may still suffer from underfitting under certain masking patterns. In such cases, given the prompt "$<\text{MASK}>bc <\text{MASK}>$", the model might incorrectly predict "$abcx$" instead of the intended sequence. This behavior can likewise induce time oscillation during sampling.

# D LIMITATIONS

## D.1 DISCUSSIONS ON LIMITATIONS

While our temporal self-consistency voting and post-training approach demonstrates effectiveness in many scenarios, it exhibits significant limitations when applied to tasks where the model's intermediate predictions are consistently inaccurate. As discussed in Appendix D.2, for the Sudoku dataset, the average correctness across all intermediate generation steps remains exceedingly low (below 5%), making it difficult to reliably vote for the correct answer. Similarly, RFT relies on the model already achieving reasonably good performance to produce meaningful reward signals (Prabhudesai et al., 2025; Agarwal et al., 2025), though combining TSE with the accuracy reward may alleviate this issue to some extent. This underscores that our approach depends on the model's inherent ability to generate correct or near-correct answers in the sampling trajectory.

Table S3: **Performance of temporal self-consistency voting and temporal consistency reinforcement on the Sudoku dataset.** The baseline model is obtained by applying supervised fine-tuning to LLaDA-8B-Instruct using the s1K dataset.

| Model | Method / Seq Len | Sudoku | | |
|---|---|---|---|---|
| | | 128 | 256 | 512 |
| | baseline | 12.2 | 6.7 | 5.5 |
| | + temporal voting | 12.5 | 6.1 | 2.8 |
| + RFT | accuracy reward (d1) | 23.2 | 17.8 | 12.7 |
| | negative TSE reward (ours) | 15.2 | 9.4 | 3.3 |
| | combining both (ours) | **27.5** | **27.8** | **16.6** |

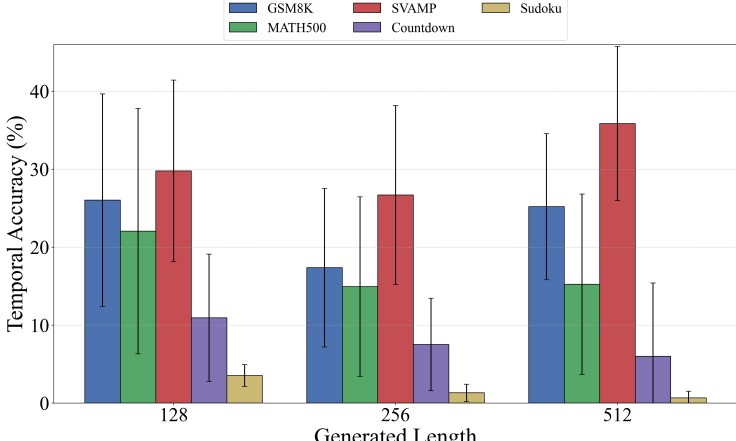

Figure S4: **Temporal accuracy across datasets.** Temporal Accuracy on Sudoku remains consistently low (all below 5%) across different settings, indicating that the model rarely generates correct answers during the sampling process. This scarcity of valid candidates severely limits the effectiveness of voting mechanisms, as there are insufficient correct outputs to reliably converge on the correct answer.

## D.2 FAILURE CASE ANALYSIS

As discussed in Appendix D.1, our method may rely on the model's initial performance to achieve further improvements. To investigate this limitation, we take the challenging Sudoku dataset as a case study. As shown in Table S3, directly applying our proposed temporal self-consistency voting and temporal consistency reinforcement with the negative TSE reward alone results in a noticeable performance drop on the Sudoku dataset. For example, the original accuracy of the base model for the generation length of 512 is 5.5%, while the voting method achieves 2.8%, reflecting a decline of 2.7%.

To better understand this problem, we conduct a deeper analysis of the model's behavior during generation. We define a metric called **Temporal Accuracy**, which quantifies the average correctness across all intermediate sampling steps in the generation process. Formally, let $e_{i,t}$ represent the correctness indicator for the $i$-th question at the $t$-th sampling step, where $e_{i,t} = 1$ if the answer is correct, and 0 otherwise. Then, the **Temporal Accuracy** for a dataset with $N$ examples and $T$ sampling steps is computed as:

$$\text{TemporalAccuracy} = \frac{1}{N \cdot T} \sum_{i=1}^{N} \sum_{t=1}^{T} e_{i,t}. \tag{S1}$$

As shown in Fig. S4, the **Temporal Accuracy** of the Sudoku is much lower than the other 4 datasets, where temporal self-consistency voting proved effective. On the Sudoku dataset, we observe that the Temporal Accuracy remains exceedingly low—below 5% on average—across all intermediate

Table S4: **Temporal self-consistency voting results on ARC-C and Winogrande using the semi-autoregressive setup (block size 32)**. Exponential and linear weighting schemes yield consistent accuracy improvements without QA-specific tuning, demonstrating robust gains across both datasets.

| | | ARC-C | | | Winogrande | | |
|---|---|---|---|---|---|---|---|
| | Method / Seq Len | 128 | 256 | 512 | 128 | 256 | 512 |
| **LLaDA-8B-Instruct** | baseline | 81.4 | 83.3 | 83.1 | 53.1 | 53.6 | 52.5 |
| | Fixed Weighting | **82.6** | **83.8** | **83.4** | 54.4 | 55.1 | 55.0 |
| + Temporal Voting | Linear Weighting | **82.6** | **83.8** | **83.4** | 56.1 | 57.7 | 58.4 |
| | Exp. Weighting | **82.6** +1.2 | **83.8** +0.5 | **83.4** +0.3 | 54.4 +1.3 | 55.1 +1.5 | 55.0 +2.5 |
| | EverPass @1 \| $t$ | 86.1 | 86.9 | 85.5 | 61.5 | 61.6 | 62.0 |

Table S5: **Temporal Consistency Reinforcement results on ARC-C and Winogrande.** Both TSE-only and combined rewards consistently outperform the baseline across sequence lengths, with the combined reward yielding the largest gains.

| | | ARC-C | | | Winogrande | | |
|---|---|---|---|---|---|---|---|
| | Method / Seq Len | 128 | 256 | 512 | 128 | 256 | 512 |
| **LLaDA** | baseline | 81.6 | 82.8 | 83.4 | 54.5 | 56.6 | 56.5 |
| + RFT | negative TSE reward (ours) | 84.5 +2.9 | 84.8 +2.0 | 84.8 +1.4 | 59.1 +4.6 | 60.6 +4.0 | 61.3 +4.8 |
| | combining both (ours) | 84.5 +2.9 | 85.8 +3.0 | 86.3 +2.9 | 66.6 +12.1 | 69.0 +12.4 | 68.5 +12.0 |

steps. This low signal makes it difficult for temporal voting mechanisms to reliably identify the correct answer and for reinforcement learning signals to guide the model effectively.

Interestingly, while RFT with only negative TSE reward leads to poorer results, combining TSE with accuracy reward can achieve better performance than using accuracy reward alone. We hypothesize that this is because the integration of TSE allows the model to receive more fine-grained rewards, rather than just binary outcomes of correct or incorrect.

# E    MORE EXPERIMENTAL RESULTS

## E.1    EXPERIMENTS ON NATURAL LANGUAGE QUESTION-ANSWERING DATASETS

To verify that our methods generalize beyond math reasoning, we added experiments on two natural language QA benchmarks, ARC-C (Clark et al., 2018) and Winogrande (Keisuke et al., 2019), evaluating both Temporal Self-Consistency Voting and Temporal Consistency Reinforcement.

**Temporal Self-Consistency Voting.** We adopt exactly the same semi-autoregressive decoding configuration as described in the main paper, using a block size of 32. For the exponential weighting scheme, we directly reuse the hyperparameter $\alpha = 1.5$, which was originally tuned on mathematical reasoning datasets, and perform no additional tuning on QA tasks. As shown in Table S4, on ARC-C, exponential temporal voting yields an average improvement of approximately +0.7 accuracy points over the baseline. On Winogrande, the untuned exponential weighting achieves an average gain of +1.8 points, and applying linear weighting leads to further improvements, reaching 58.4 accuracy (compared to a 52.5 baseline at length 512). These findings demonstrate that temporal voting provides consistent benefits on natural-language QA tasks and remains robust under different weighting schemes.

**Temporal Consistency Reinforcement.** We train on ARC-C and Winogrande using the same GRPO configuration as in the main experiments, adopting either (i) a negative TSE reward alone or (ii) a combined TSE + accuracy reward. As shown in Table S7, both configurations consistently outperform the baseline across all sequence lengths on ARC-C, yielding improvements of approximately 2–3 accuracy points (e.g., 81.6 to 84.5 at 128 tokens and 82.8 to 85.8 at 256

Table S6: **Unified Model Performance Across Multiple Tasks.** A single model is trained jointly on GSM8K, MATH500, SVAMP, and Countdown.

| | Method / Seq Len | GSM8K | | | MATH500 | | | SVAMP | | | Countdown | | |
|---|---|---|---|---|---|---|---|---|---|---|---|---|---|
| | | 128 | 256 | 512 | 128 | 256 | 512 | 128 | 256 | 512 | 128 | 256 | 512 |
| **LLaDA** | baseline | 70.2 | 78.7 | 80.1 | 23.8 | 34.4 | 36.8 | 81.7 | 83.3 | 82.7 | 21.5 | 19.9 | 21.5 |
| | accuracy reward (d1) | 71.3 | 77.3 | 80.7 | 28.0 | 33.6 | 39.2 | 84.0 | 84.0 | 85.0 | 25.8 | 22.3 | 37.1 |
| + RFT | negative TSE reward (ours) | 71.5 | 77.2 | **81.2** | **29.0** | 33.8 | 39.0 | 86.3 | 84.7 | **87.0** | 36.7 | 25.4 | 46.1 |
| | combined reward (ours) | **72.4** | **78.8** | 80.4 | 28.8 | **35.2** | **40.2** | **88.7** | **87.0** | 86.0 | **37.1** | **27.7** | **46.9** |
| **LLaDA-1.5** | baseline | 69.8 | 79.4 | 81.1 | 29.0 | 32.4 | 35.4 | 85.3 | 86.3 | 83.3 | 21.5 | 21.1 | 20.7 |
| | accuracy reward (d1) | 72.5 | 79.2 | 81.7 | 28.6 | **35.0** | 39.0 | 89.0 | 87.7 | 85.3 | 30.5 | 24.6 | 41.8 |
| + RFT | negative TSE reward (ours) | 72.5 | 79.2 | 81.4 | **29.0** | 32.6 | 38.2 | **89.3** | 85.3 | 86.7 | 37.3 | **44.5** | 34.0 |
| | combined reward (ours) | **73.7** | **79.5** | **82.5** | 28.2 | **35.0** | **41.6** | 88.3 | **90.0** | **88.0** | **37.9** | 29.7 | **50.4** |

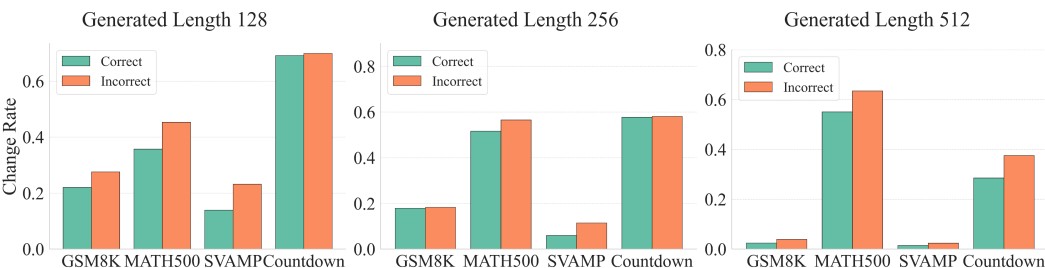

Figure S5: **ChangeRate of correct answers across different output lengths for each dataset.** Lower ChangeRate indicates more stable predictions over the sampling trajectory.

tokens). On Winogrande, the combined reward yields the largest gains, with improvements of up to +12.4 points (e.g., 56.6 to 69.0 at 256 tokens).

Overall, these additional experiments on ARC-C and Winogrande demonstrate that both our test-time voting and reinforcement learning approaches generalize beyond mathematical reasoning to natural-language QA. This further supports our claim that leveraging temporal dynamics constitutes a broadly applicable principle for improving diffusion language models.

### E.2 TRAINING A UNIFIED MODEL ACROSS MULTIPLE TASKS

To evaluate the generalization ability of our approach, we validate the proposed method in a multi-task setting. Specifically, we jointly train a single model on four datasets: GSM8K, MATH500, SVAMP, and Countdown. To ensure balanced training across tasks, we subsample each dataset so that the number of training examples is equal for all tasks.

Table S6 summarizes the results. Across all four benchmarks, our combined reward method consistently outperforms the supervised fine-tuning baselines. On GSM8K, we observe clear gains at shorter sequence lengths, for example, with LLaDA-1.5 on length 128, accuracy improves from 69.8 to 73.7, yielding +3.9 points. On SVAMP, similar short-sequence gains are observed, from 81.7 to 88.7, a +7.0 points increase. For the more challenging MATH500, the best improvement occurs at LLaDA-1.5, length 512, where performance increases from 35.4 to 41.6 (+6.2 points). Finally, on Countdown, our method achieves dramatic improvements, for instance, with LLaDA-1.5 at length 512, accuracy climbs from 20.7 to 50.4, a striking +29.7 points.

These results demonstrate that our method not only scales well across different reasoning domains but also enhances robustness under varying sequence lengths, confirming its effectiveness as a unified reward design for multi-task learning.

Table S7: Ablation results replacing TSE with alternative metrics, including negative average token entropy (ATE) and pairwise agreement (PA) between consecutive diffusion steps. **Bolded** values indicate the best performance among the three methods.

| Method / Seq Len | GSM8K | | | MATH500 | | | SVAMP | | | Countdown | | |
|---|---|---|---|---|---|---|---|---|---|---|---|---|
| | 128 | 256 | 512 | 128 | 256 | 512 | 128 | 256 | 512 | 128 | 256 | 512 |
| baseline | 70.2 | 78.7 | 80.1 | 23.8 | 34.4 | 36.8 | 81.7 | 83.3 | 82.7 | 21.5 | 19.9 | 21.5 |
| negative TSE reward | **72.2** | 78.8 | **80.2** | 30.6 | **34.6** | **38.0** | 84.3 | 89.0 | 88.7 | 38.6 | 53.5 | 44.9 |
| negative ATE reward | 69.1 | 77.7 | 79.5 | 27.4 | 32.2 | 37.2 | 82.7 | 85.3 | 85.0 | 22.9 | 23.4 | 30.5 |
| PA reward | 71.3 | **78.9** | 79.7 | **31.0** | 33.0 | 37.6 | 83.0 | 87.3 | 86.7 | 32.0 | 44.7 | 41.3 |
| TSE & Accuracy reward | **72.1** | **80.0** | **83.0** | 31.2 | **35.4** | **41.4** | **85.0** | **90.3** | **92.3** | **41.5** | 42.6 | **54.7** |
| ATE & Accuracy reward | 70.2 | 77.2 | 80.2 | 30.8 | 34.8 | 38.4 | 84.7 | 86.3 | 85.3 | 28.1 | 29.7 | 35.9 |
| PA & Accuracy reward | 71.0 | 79.2 | 81.4 | **32.0** | 33.8 | 39.6 | **85.0** | 89.7 | 88.7 | 36.9 | **43.4** | 50.5 |

### E.3 POSSIBLE ALTERNATIVES TO TEMPORAL SEMANTIC ENTROPY

Our method primarily adopts Temperal Semantic Entropy(TSE) as the underlying mechanism for temporal consistency modeling. However, TSE is not the only viable choice. To contextualize our design and clarify the landscape of alternatives, we examine two additional possible approaches: pairwise agreement and token entropy.

To measure temporal consistency, a natural idea is to quantify how frequently the predicted answer changes along the sampling trajectory. Motivated by this intuition, we introduce $\mathrm{ChangeRate}$ as a metric for assessing the model's confidence and stability.

During decoding, we obtain a sequence of T intermediate predictions, denoted as $\{\mathbf{x}_0^t\}_{t=1}^T$. Based on this sequence, we define $\mathrm{ChangeRate}$ as the fraction of consecutive steps that produce the different answer:

$$\mathrm{ChangeRate} = \frac{1}{T-1} \sum_{t=1}^{T-1} \mathbf{1}[\mathbf{x}_0^t = \mathbf{x}_0^{t+1}] \tag{S2}$$

A higher $\mathrm{ChangeRate}$ indicates that the model's predictions vary more frequently across successive steps, signaling lower temporal consistency. For completeness, we define pairwise agreement (PA) as the complement of ChangeRate, $\mathrm{PA} = 1 - \mathrm{ChangeRate}$, and employ it as an RL reward to promote temporally consistent predictions.

Similar to TSE, we compute the $\mathrm{ChangeRate}$ for correct answers across different output lengths, with the results shown in Fig. S5. We observe that, for correctly answered questions, the $\mathrm{ChangeRate}$ tends to be lower. However, unlike TSE, $\mathrm{ChangeRate}$ does not exhibit significant variation across datasets. That is, whether the model performs well or poorly on a given dataset, the $\mathrm{ChangeRate}$ does not show a clear pattern. In contrast, TSE is noticeably lower on easier datasets such as GSM8K and SVAMP compared to more challenging ones like MATH500 and Countdown, as shown in Sec. 3.3.

In addition, we propose using the answer entropy, which is computed from the token distribution of the predicted answer, as a supervisory signal. Although this entropy measure cannot directly supervise temporal consistency in the answer trajectory, it still provides a meaningful indication of how confident the model is in its prediction. As shown in Fig. 3, higher entropy corresponds to lower confidence, while lower entropy suggests that the model is more certain about the generated answer.

As shown in Table S7, we observe that: (1) Using negative token entropy as the reward yields moderate improvement, indicating that local confidence is useful but insufficient. Gains are consistently smaller than with TSE; (2) The pairwise agreement reward performs better than token entropy and comparable to TSE.

Intuitively, PA captures dynamic stability; for example, it can tell that "010101" is less stable than "000111," but it cannot differentiate between semantically diverse outputs such as "010203" and "010101." In contrast, TSE measures the semantic dispersion of intermediate predictions, capturing

Table S8: **Ablations on scoring rules for combining TSE with accuracy reward.** We compare 4 different scoring rules, including entropy scoring, quadratic scoring, spherical scoring, and logistic scoring.

| Model / Dataset | | GSM8K | | | MATH500 | | | SVAMP | | | Countdown | | |
|---|---|---|---|---|---|---|---|---|---|---|---|---|---|
| | Method / Seq Len | 128 | 256 | 512 | 128 | 256 | 512 | 128 | 256 | 512 | 128 | 256 | 512 |
| **LLaDA** | baseline | 70.2 | 78.7 | 80.1 | 23.8 | 34.4 | 36.8 | 81.7 | 83.3 | 82.7 | 21.5 | 19.9 | 21.5 |
| +RFT | entropy | 71.7 | 78.5 | 82.3 | 31.6 | **38.2** | 39.2 | **88.7** | 89.3 | 89.3 | 47.6 | **50.0** | 53.1 |
| | quadratic | 71.3 | 79.9 | 82.0 | 31.0 | 37.6 | 40.0 | 88.0 | 8.3 | 89.3 | **50.0** | 34.4 | 48.1 |
| | logistic | **73.0** | 79.5 | 81.1 | **31.6** | 36.8 | 39.4 | 87.7 | 87.3 | 90.7 | 46.5 | 37.9 | 51.2 |
| | spherical | 72.1 | **80.0** | **83.0** | 31.2 | 35.4 | **41.4** | 85.0 | **90.3** | **92.3** | 41.5 | 42.6 | **54.7** |
| **LLaDA-1.5** | baseline | 69.8 | 79.4 | 81.1 | 29.0 | 32.4 | 35.4 | 85.3 | 86.3 | 83.3 | 21.5 | 21.1 | 20.7 |
| +RFT | entropy | 71.2 | 79.2 | 83.4 | 27.8 | **36.6** | 41.2 | 86.7 | **90.3** | 89.0 | 44.1 | 45.3 | 58.6 |
| | quadratic | **73.4** | 79.1 | 83.2 | 29.2 | 33.2 | **42.4** | 86.0 | 86.7 | 90.0 | 43.4 | 45.3 | 57.4 |
| | logistic | 71.2 | 79.2 | 83.1 | **30.2** | 33.8 | 41.0 | **88.3** | 89.0 | **91.0** | 48.8 | **46.9** | 61.3 |
| | spherical | 73.2 | **80.5** | **84.0** | 29.6 | 35.4 | 41.4 | 86.3 | **90.3** | 89.0 | 44.5 | **46.9** | **63.3** |

how the meaning of partial answers drifts along the trajectory, but it is less sensitive to fine-grained dynamical fluctuations among semantically similar states.

We believe an ideal temporal-consistency metric should integrate the complementary strengths of PA and TSE, leveraging PA's sensitivity to dynamics and TSE's sensitivity to semantic variation, and we plan to explore such hybrid rewards in future work.

### E.4 ABLATIONS ON SCORING RULES FOR COMBINING TSE WITH ACCURACY REWARD

In Sec. 4.2, we combine TSE with the accuracy reward using the proper scoring rule (Gneiting & Raftery, 2007). The purpose of these scoring rules is to promote truthful confidence assessments: they attain their minimum value when the predicted confidence $c$ precisely mirrors the actual probability that the model's output $o$ corresponds to the correct answer $o^*$. An exception is the spherical version we use, which consistently encourages both correctness and higher certainty, without penalizing overconfidence in incorrect predictions. Here, we conduct an ablation study on different scoring rules. Specifically, we consider the following four forms:

$$\textbf{Entropy scoring:} \quad r_i^{\text{ent}} = \mathbb{1}_{o_i=o^*} \cdot c(o_i)$$

$$\textbf{Quadratic scoring:} \quad r_i^{\text{quad}} = \mathbb{1}_{o_i=o^*} - \left(c(o_i) - \mathbb{1}_{o_i=o^*}\right)^2$$

$$\textbf{Logistic scoring:} \quad r_i^{\text{log}} = \mathbb{1}_{o_i=o^*} + \mathbb{1}_{o_i=o^*} \log(c(o_i)) + (1 - \mathbb{1}_{o_i=o^*}) \log(1 - c(o_i))$$

$$\textbf{Spherical scoring:} \quad r_i^{\text{sphe}} = \mathbb{1}_{o_i=o^*} + \frac{c(o_i)}{\sqrt{(c(o_i))^2 + (1 - c(o_i))^2}}$$

All four proposed reward functions aim to jointly encourage correctness and temporal self-consistency. Notably, the $r_i^{\text{ent}}$ function gives a reward of 0 for incorrect answers, while for correct answers, the reward is given by $c(o_i) = \frac{\mathcal{H}_{\max} - \text{TSE}(o_i)}{\mathcal{H}_{\max}}$, where $\mathcal{H}_{\max} = \log T$. This reward reaches a maximum value of 1 when all sampling steps yield the correct answer. The remaining three functions, $r_i^{\text{quad}}$, $r_i^{\text{log}}$, and $r_i^{\text{sphe}}$, correspond to the commonly used quadratic scoring, logarithmic scoring, and spherical scoring in proper scoring rules (Gneiting & Raftery, 2007), respectively. We report the performance of all four reward combination methods in Table S8. By default, we use the spherical scoring rule because it demonstrates more superior results compared to the alternatives.

### E.5 TEMPORAL SELF-CONSISTENCY VOTING AFTER RFT

It is worthwhile to investigate whether Temporal Self-consistency Voting continues to provide performance benefits after Temporal Consistency Reinforcement. To this end, we conducted experiments using two models: the first is derived from the LLaDA-8B-Instruct model trained with the negative TSE reward, and the second is trained using the accuracy reward combined with TSE.

Table S9: **Performance of temporal majority voting after reinforcement learning.** Temporal self-consistency voting was applied to the model fine-tuned via temporal consistency reinforcement. The upper part is derived from the LLaDA-8B-Instruct model trained using the Negative TSE reward, whereas the lower part is based on the model trained with a combination of TSE and accuracy reward. For reference, we include the oracle EverPass@1 $\mid t$ as a performance upper bound.

| | | GSM8K | | | MATH500 | | | SVAMP | | | Countdown | | |
|---|---|---|---|---|---|---|---|---|---|---|---|---|---|
| | Method / Seq Len | 128 | 256 | 512 | 128 | 256 | 512 | 128 | 256 | 512 | 128 | 256 | 512 |
| **Negative TSE reward** | After RFT | 72.2 | 78.8 | 80.2 | 30.6 | 34.6 | 38.0 | 84.3 | 89.0 | 88.7 | 38.6 | 53.5 | 44.9 |
| + Temporal Voting | Fixed Weighting | 70.4 | 77.6 | 80.2 | 30.8 | 34.4 | 37.6 | 85.0 | 88.3 | 88.7 | 39.5 | 53.5 | 45.7 |
| | Linear Weighting | 72.3 | 78.8 | 80.6 | 30.8 | 35.4 | 38.0 | 85.3 | 88.3 | 88.7 | 39.1 | 53.5 | 45.7 |
| | Exp. Weighting | 72.6
+0.4 | 79.2
+0.4 | 80.6
+0.4 | 30.8
+0.2 | 35.0
+0.4 | 38.0
+0.0 | 85.3
+1.0 | 89.0
+0.0 | 88.7
+0.0 | 39.5
+0.9 | 53.9
+0.4 | 45.7
+0.8 |
| | EverPass@1 $\mid t$ | 80.5 | 81.8 | 81.3 | 33.4 | 37.4 | 40.0 | 88.7 | 90.7 | 89.7 | 44.1 | 59.8 | 55.5 |
| **TSE & accuracy reward** | After RFT | 72.1 | 80.0 | 83.0 | 31.2 | 35.4 | 41.4 | 85.0 | 90.3 | 92.3 | 41.5 | 42.6 | 54.7 |
| + Temporal Voting | Fixed Weighting | 71.2 | 79.6 | 82.8 | 31.2 | 35.6 | 41.2 | 85.3 | 90.7 | 92.3 | 40.6 | 41.5 | 53.2 |
| | Linear Weighting | 73.0 | 81.9 | 83.0 | 31.2 | 36.0 | 41.0 | 86.7 | 90.3 | 92.7 | 40.9 | 42.8 | 54.2 |
| | Exp. Weighting | 72.6
+0.5 | 81.1
+1.1 | 83.3
+0.3 | 31.6
+0.4 | 35.8
+0.4 | 41.4
+0.0 | 86.3
+1.3 | 90.7
+0.4 | 92.3
+0.0 | 41.5
+0.0 | 42.7
+0.1 | 55.1
+0.4 |
| | EverPass@1 $\mid t$ | 84.0 | 91.4 | 87.9 | 43.6 | 49.2 | 52.0 | 90.7 | 91.7 | 92.7 | 54.3 | 68.0 | 70.3 |

We applied Temporal Self-Consistency Voting to both models, and the results are summarized in Table S9. The findings indicate that both models still exhibit performance improvements when temporal voting is applied, even after undergoing reinforcement learning. This suggests that Temporal Self-Consistency Voting and Temporal Consistency Reinforcement are complementary techniques, and can be effectively combined to further enhance model performance.

### E.6 EXAMPLES OF TIME OSCILLATION

We present representative examples of temporal oscillation from GSM8K, using LLaDA-8B-Instruct as the backbone model. Note that all examples shown below were ultimately incorrect but were classified as correct answers by our voting method, as described in Sec. 4.1. Correct answers are in green, incorrect answers in red. We use blue to highlight the key segments in each example.

We observe that many questions become incorrect during temporal oscillation due to flawed reasoning paths. Although the model may initially provide the correct answer and follow a logical reasoning path, an incorrect trajectory can ultimately lead to an erroneous response.

A notable case is seen in Example 3. Initially, the model correctly calculates the total cups, per-hour revenue, and the per-hour cost deduction to derive the right profit per hour (step 54). However, during reasoning, it introduces erroneous masking regarding key calculation elements—the per-hour cost to subtract. Even though the answer remains correct at first (step 61), the flawed reasoning trajectory eventually results in miscalculations and an incorrect final response (step 62).

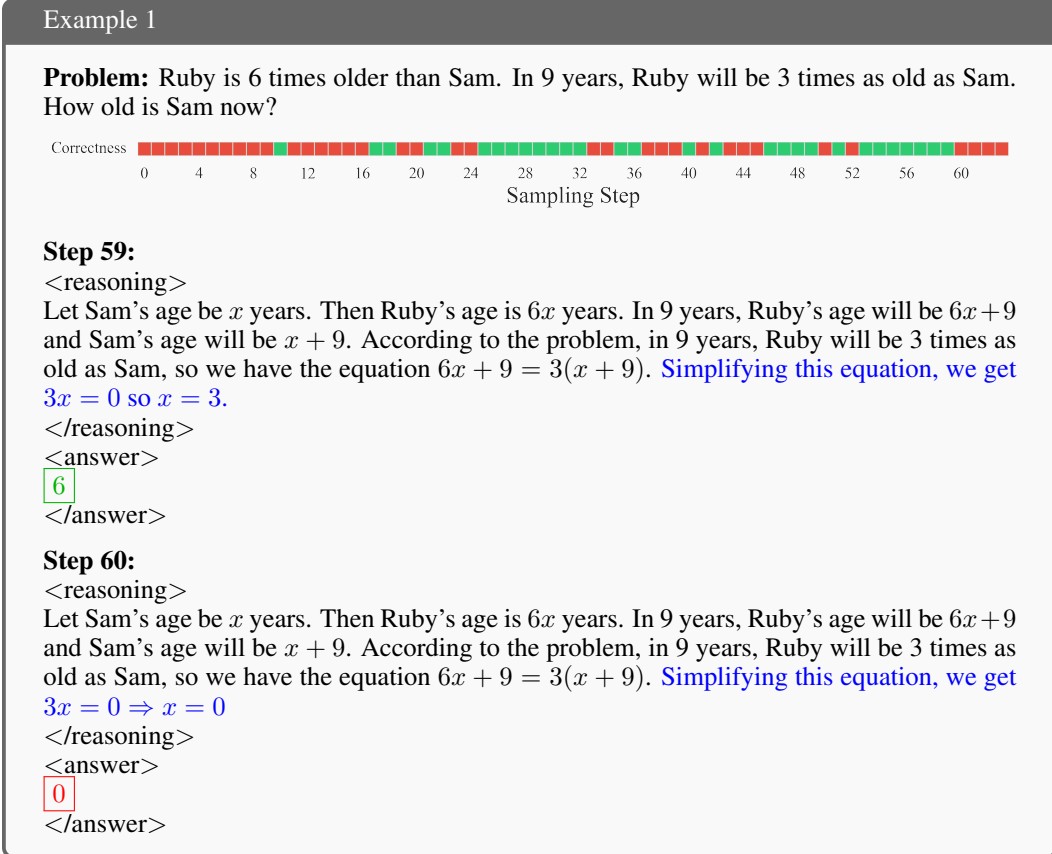

**Example 1**

**Problem:** Ruby is 6 times older than Sam. In 9 years, Ruby will be 3 times as old as Sam. How old is Sam now?

**Step 59:**

$<$reasoning$>$

Let Sam's age be $x$ years. Then Ruby's age is $6x$ years. In 9 years, Ruby's age will be $6x+9$ and Sam's age will be $x+9$. According to the problem, in 9 years, Ruby will be 3 times as old as Sam, so we have the equation $6x+9 = 3(x+9)$. Simplifying this equation, we get $3x = 0$ so $x = 3$.

$<$/reasoning$>$

$<$answer$>$

6

$<$/answer$>$

**Step 60:**

$<$reasoning$>$

Let Sam's age be $x$ years. Then Ruby's age is $6x$ years. In 9 years, Ruby's age will be $6x+9$ and Sam's age will be $x+9$. According to the problem, in 9 years, Ruby will be 3 times as old as Sam, so we have the equation $6x+9 = 3(x+9)$. Simplifying this equation, we get $3x = 0 \Rightarrow x = 0$

$<$/reasoning$>$

$<$answer$>$

0

$<$/answer$>$

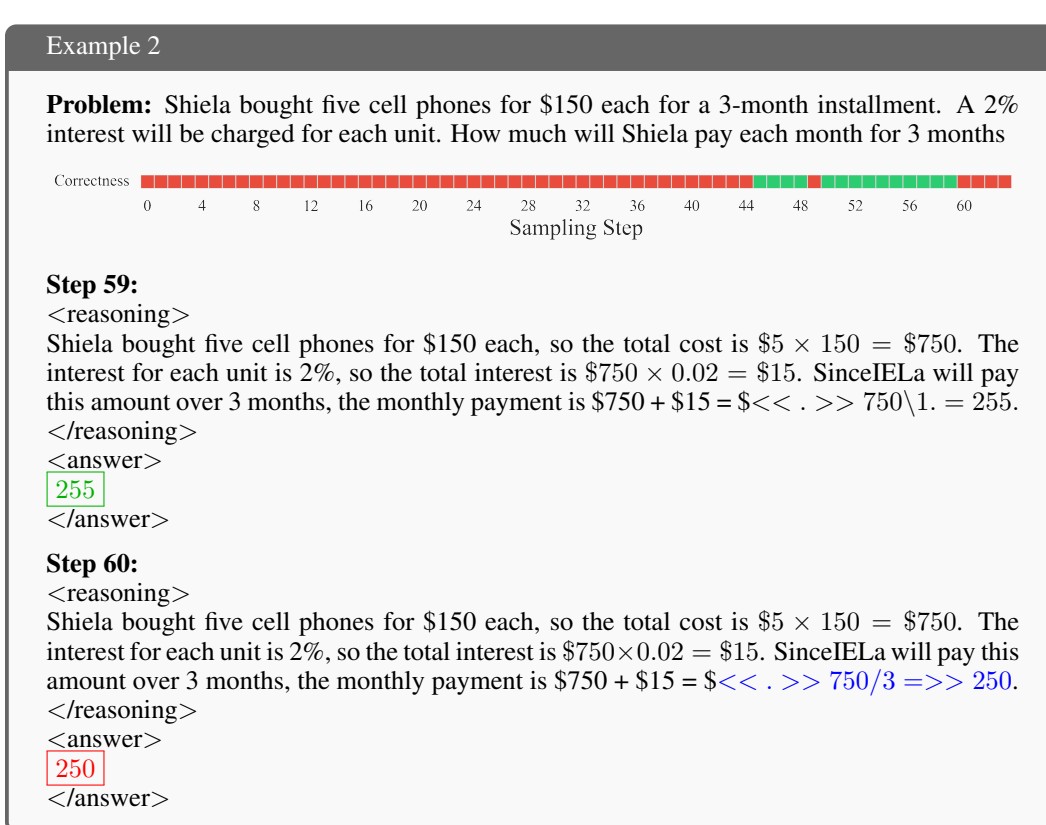

**Example 2**

**Problem:** Shiela bought five cell phones for $150 each for a 3-month installment. A 2% interest will be charged for each unit. How much will Shiela pay each month for 3 months

**Step 59:**

$<$reasoning$>$

Shiela bought five cell phones for $150 each, so the total cost is $\$5 \times 150 = \$750$. The interest for each unit is 2%, so the total interest is $\$750 \times 0.02 = \$15$. SinceIELa will pay this amount over 3 months, the monthly payment is $\$750 + \$15 = \$<<\ .\ >> 750\backslash 1. = 255$.

$<$/reasoning$>$

$<$answer$>$

255

$<$/answer$>$

**Step 60:**

$<$reasoning$>$

Shiela bought five cell phones for $150 each, so the total cost is $\$5 \times 150 = \$750$. The interest for each unit is 2%, so the total interest is $\$750 \times 0.02 = \$15$. SinceIELa will pay this amount over 3 months, the monthly payment is $\$750 + \$15 = \$<<\ .\ >> 750/3 =>> 250$.

$<$/reasoning$>$

$<$answer$>$

250

$<$/answer$>$

## Example 3

**Problem:** Millie decides to open a lemonade stand. She spends $18 to buy enough supplies to make 3 pitchers of lemonade. Each pitcher holds 12 cups of lemonade. She sells each cup of lemonade for $1. She sells an average of 4 cups per hour that her lemonade stand is open. If Millie sells all of the lemonade, how much profit will she make per hour that she spends running the lemonade stand?

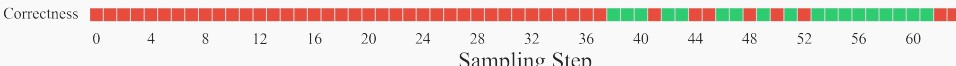

**Step 54:**
<reasoning>
Millie spends $18 on supplies to make 3 pitchers of lemonade. Each pitcher holds 12 cups, so she has a total of 3 * 12 = 36 cups of lemonade. She sells each cup for $1 and and sells an average of 4 cups per hour. Therefore, her revenue per hour is 4 * $1 = $4. To find per hour profit, we subtract the cost of supplies from the revenue: $4 - $2 = $2 per hour.
</reasoning>
<answer>
2
</answer>

. . .

**Step 61:**
<reasoning>
Millie spends $18 on supplies to make 3 pitchers of lemonade. Each pitcher holds 12 cups, so she has a total of 3 * 12 = 36 cups of lemonade. She sells each cup for $1 and and sells an average of 4 cups per hour. Therefore, her revenue per hour is 4 * $1 = $4. To find per hour profit, we subtract the cost of supplies from the revenue: $4 - $3 = $2 per hour.
</reasoning>
<answer>
2
</answer>

**Step 62:**
<reasoning>
Millie spends $18 on supplies to make 3 pitchers of lemonade. Each pitcher holds 12 cups, so she has a total of 3 * 12 = 36 cups of lemonade. She sells each cup for $1 and and sells an average of 4 cups per hour. Therefore, her revenue per hour is 4 * $1 = $4. To find per hour profit, we subtract the cost of supplies from the revenue: $4 - $3 = $1 per hour.
</reasoning>
<answer>
1
</answer>

