# OpenReview forum: "Time Is a Feature: Exploiting Temporal Dynamics in Diffusion Language Models"
_ICLR.cc/2026/Conference — ICLR 2026 Poster_

### Official Review · Reviewer_39aQ · 2025-10-26

**Soundness:** 3
**Presentation:** 3
**Contribution:** 4
**Rating:** 8
**Confidence:** 3

**Summary:**

The paper addresses temporal oscillation in dLLMs, where correct answers often appear during intermediate denoising steps but are later overwritten. Building on insights from self-consistency and semantic entropy, the authors propose two complementary approaches: (1) Temporal Self-Consistency Voting, a training-free decoding method that aggregates intermediate predictions, and (2) Temporal Consistency Reinforcement, a post-training method using GRPO with Temporal Semantic Entropy as a reward to promote stable generations. Experiments on mathematical reasoning benchmarks demonstrate consistent improvements, with particularly strong gains when TSE is combined with accuracy-based rewards.

**Strengths:**

- The framing of temporal oscillation as a central issue in dLLM decoding and devising a way of extracting signal from the intermediate outputs is both novel and timely given the emerging importance of dLLMs.
- This work presents strong empirical evidence, including ablation studies on reward design and weighting strategies.
- Newly introduced metrics, such as EverPass and TSE, are intuitive and well-motivated, with their significance demonstrated through clear and quantitative analysis.

**Weaknesses:**

- While the evaluation spans four datasets (GSM8K, MATH500, SVAMP, Countdown), this focuses primarily on mathematical reasoning tasks. Extending the analysis to other domains (e.g., natural language QA) would strengthen the case for the generalizability of this approach.
- The success of the introduced strategies, inference-time voting as well as RL fine-tuning, appears to rely heavily on the baseline model's accuracy, which may suggest limited gains for harder reasoning tasks to expand the applicability of dLLMs broadly (although this point is mentioned in Appendix D.1).

I also note the existence of a concurrent work (https://arxiv.org/abs/2508.19982) on closely related problem and method, while it assumes early stabilization of the prediction and proposes stopping criteria for training-free approach. This is not considered in this review, since they are concurrent works.

**Questions:**

- Since temporal oscillation arises from the combination of remasking and incorrect denoising, it could be in principle be mitigated by a modified diffusion schedule. How much of the observed oscillation is inherent to dLLMs (and thus requires methods like those proposed here), versus being reducible by scheduling or inference design choices?
- The use of TSE in this work is empirically motivated, and it would be possible to consider other types of temporal consistency metrics or rewards, for example, averaging the pairwise agreement between consecutive timesteps of a (later) denoising trajectory. What is the empirical advantage of using entropy-based methods over such alternatives?

Minor typos/comments
- p. 8 Paragraph "Ablations on Voting Strategies." Fig. 5b -> Fig. 5a
- p. 4 Eq (1). The use of $x_0$ on the RHS is inconsistent with the LHS. Given the later definition of $x_0^t$, it would be more precise to write $\sum_{x_0^t} q(x_{t-1}|x_0^t) p_\theta (x_0^t | x_t)$? Also, the later decoding step would be $x_0^t \sim p_\theta(x_0^t | x_t)$.

---

> ### Author Response · Authors · 2025-11-20
> **Response to Reviewer 39aQ (Part 1)**
>
> # W1: Extending the analysis to other domains (e.g., natural language QA) would strengthen the case for the generalizability of this approach.
>
> Thank you for the valuable suggestion. To verify that our methods generalize beyond math reasoning, we added experiments on two natural language QA benchmarks, **ARC-C** and **Winogrande**, evaluating both **Temporal Self-Consistency Voting** and **Temporal Consistency Reinforcement**.
>
> **(1)  Temporal Self-Consistency Voting**
> We follow exactly the same **semi-autoregressive** decoding setup as in the main paper (block size 32).  For exponential weighting we reuse the hyperparameter **α = 1.5** tuned on math datasets, without any tuning on QA. The main voting results is shown below:
>
> | Method / Seq Len | ARC-C-128 | ARC-C-256 | ARC-C-512 | Winogrande-128 | Winogrande-256 | Winogrande-512 |
> |------------------|------|------|------|------|------|------|
> | **Baseline**     | 81.4 | 83.3 | 83.1 | 53.1 | 53.6 | 52.5 |
> | **Fixed Weighting** | 82.6 | 83.8 | 83.4 | 54.4 | 55.1 | 55.0 |
> | **Linear Weighting** | 82.6 | 83.8 | 83.4 | 56.1 | 57.7 | 58.4 |
> | **Exp. Weighting (α=1.5)** | 82.6 | 83.8 | 83.4 | 54.4 | 55.1 | 55.0 |
> | **Δ (Exp − Base)** | +1.2 | +0.5 | +0.3 | +1.3 | +1.5 | +2.5 |
> | **EverPass** | 86.1 | 86.9 | 85.5 | 61.5 | 61.6 | 62.0 |
>
> On **ARC-C**, exponential temporal voting improves accuracy by about **+0.7 points on average** over the baseline. On **Winogrande**, the untuned exponential weighting yields about **+1.8 points on average**, and **linear weighting** further improves performance, reaching up to **58.4** (vs. **52.5** baseline at length 512). These results indicate that temporal voting yields consistent gains on natural language QA and is robust to the choice of weighting scheme.
>
> **(2)  Temporal Consistency Reinforcement**
> For **Temporal Consistency Reinforcement**, we train on ARC-C and Winogrande with the same GRPO setup as in the main experiments, using either (i) a **negative TSE reward** alone or (ii) a **combined TSE + accuracy reward**. Both configurations consistently outperform the baseline across all sequence lengths on ARC-C, yielding gains of about 2–3 points (e.g., from 81.6→84.5 at 128 tokens and 82.8→85.8 at 256 tokens), while on Winogrande the combined reward provides the largest improvements, with up to +12.4 points (e.g., 56.6→69.0 at 256 tokens). The numbers are detailed in the following table.
>
>
> | Method / Seq Len          | ARC-C-128 | ARC-C-256 | ARC-C-512 | Winogrande-128 | Winogrande-256 | Winogrande-512 |
> |---------------------------|-----------|-----------|-----------|----------------|----------------|----------------|
> | **Baseline**              | 81.6      | 82.8      | 83.4      | 54.5           | 56.6           | 56.5           |
> | **TSE only**              | 84.5      | 84.8      | 84.8      | 59.1           | 60.6           | 61.3           |
> | **Δ (TSE − Base)**        | +2.9      | +2.0      | +1.4      | +4.6           | +4.0           | +4.8           |
> | **Combined**              | 84.5      | 85.8      | 86.3      | 66.6           | 69.0           | 68.5           |
> | **Δ (Combined − Base)**   | +2.9      | +3.0      | +2.9      | +12.1          | +12.4          | +12.0          |
>
>
> Overall, the additional experiments on **ARC-C** and **Winogrande** demonstrate that both our test-time voting and reinforcement learning methods **generalize beyond math reasoning to natural language QA**, supporting our claim that exploiting temporal dynamics is a broadly applicable principle for improving diffusion language models. We have incorporated these results into **Sec. E.1** of the revised manuscript.

---

> > ### Author Response · Authors · 2025-11-20
> > **Response to Reviewer 39aQ (Part 2)**
> >
> > # W2: Reliance on baseline model's accuracy
> >
> > Thank you for the comment. We agree with the reviewer and have discussed this limitation in the manuscript, noting that temporal voting requires the baseline model to produce at least some correct intermediate states, which is not the case for tasks like Sudoku.
> >
> > Our additional experiments in Appendix D (Table S3) show that **combining TSE with the accuracy reward during RFT helps mitigate this dependency**. Even when intermediate correctness is extremely low, the combined reward improves performance over accuracy-only training, indicating that TSE provides a complementary and stabilizing learning signal.
> >
> > In summary, while temporal voting relies on baseline model competence, the RFT variant with the combined reward substantially reduces this reliance.
> >
> > ---
> >
> > # W3: Discussion with concurrent work
> >
> > Thank you for the suggestion. We have added a focused discussion of concurrent work on temporal behaviors in diffusion language models and clarified how our approach complements these studies. The manuscript has been updated accordingly, as shown below.
> >
> > Several concurrent work also investigates temporal behaviors in diffusion language models. Li et al. (2025a) [1] show that dLLMs often generate correct intermediate answers that are later overwritten, and propose stopping generation early when such signals appear. He et al. (2025) [2] further demonstrate that temporal cues along the denoising trajectory can be directly exploited to improve reasoning. Xie et al. (2025) [3] introduce a step-aware RL method that aligns denoising steps with hierarchical reasoning to avoid unstructured refinement.
> > While these studies highlight the value of early-step signals or step-specific structure, our work takes a complementary direction by developing both a lightweight test-time voting method and an unsupervised reinforcement signal grounded in trajectory-level temporal consistency.
> >
> > [1] *Diffusion Language Models Know the Answer Before Decoding*
> > [2] *MDPO: Overcoming the Training-Inference Divide of Masked Diffusion Language Model*
> > [3] *Step-Aware Policy Optimization for Reasoning in Diffusion Large Language Models*

---

> > > ### Author Response · Authors · 2025-11-20
> > > **Response to Reviewer 39aQ (Part 3)**
> > >
> > > # Q1: How much of the observed oscillation is inherent to dLLMs (and thus requires methods like those proposed here), versus being reducible by scheduling or inference design choices?
> > >
> > >
> > > Thank you for raising this important point. In the revised manuscript, we provide systematic and controlled experiments to validate that temporal oscillation is a robust and model-intrinsic phenomenon, rather than an artifact of a particular decoding configuration.
> > >
> > >
> > >
> > > | Seed | Temp. | Remasking | GSM8K 128 | GSM8K 256 | GSM8K 512 | MATH 128 | MATH 256 | MATH 512 | SVAMP 128 | SVAMP 256 | SVAMP 512 | Count. 128 | Count. 256 | Count. 512 |
> > > |------|--------|-----------|------------|------------|------------|-----------|-----------|-----------|-------------|-------------|-------------|--------------|--------------|--------------|
> > > | 42 | 0.0 | LowConf | 67.9 / 80.3 | 76.5 / 84.8 | 79.2 / 81.0 | 27.4 / 40.2 | 33.4 / 45.6 | 35.8 / 47.2 | 83.3 / 90.7 | 84.0 / 89.7 | 84.7 / 87.3 | 18.8 / 25.8 | 20.3 / 27.7 | 15.6 / 19.9 |
> > > | 218 | 0.0 | LowConf | 68.2 / 81.5 | 76.5 / 84.8 | 79.2 / 81.0 | 27.2 / 40.8 | 33.4 / 45.6 | 35.8 / 47.2 | 83.0 / 91.0 | 84.0 / 89.7 | 84.7 / 87.3 | 20.3 / 29.3 | 20.3 / 27.0 | 16.0 / 20.7 |
> > > | 66 | 0.0 | LowConf | 69.6 / 80.3 | 76.3 / 85.2 | 78.2 / 80.3 | 27.2 / 40.8 | 33.4 / 45.6 | 35.8 / 47.2 | 83.3 / 90.7 | 83.3 / 89.3 | 84.7 / 86.7 | 20.3 / 27.7 | 22.3 / 29.3 | 18.8 / 22.7 |
> > > | 42 | 0.6 | LowConf | 67.4 / 82.6 | 76.6 / 86.7 | 78.5 / 81.5 | 25.6 / 43.2 | 33.8 / 49.0 | 33.8 / 50.8 | 84.7 / 91.7 | 85.0 / 90.7 | 84.7 / 87.0 | 20.3 / 29.3 | 22.3 / 20.7 | 18.8 / 22.7 |
> > > | 42 | 1.0 | LowConf | 66.1 / 78.8 | 75.2 / 81.9 | 77.0 / 79.2 | 23.8 / 37.6 | 30.6 / 42.0 | 30.8 / 42.2 | 85.0 / 89.7 | 84.3 / 87.0 | 85.0 / 87.3 | 19.5 / 22.3 | 17.6 / 19.5 | 16.8 / 21.1 |
> > > | 42 | 0.0 | Random | 57.2 / 76.9 | 64.5 / 76.6 | 62.1 / 66.8 | 19.8 / 35.8 | 24.4 / 38.6 | 23.4 / 33.4 | 73.0 / 86.0 | 73.3 / 82.0 | 72.7 / 78.0 | 10.9 / 22.3 | 8.2 / 12.5 | 10.5 / 14.5 |
> > > | 218 | 0.0 | Random | 59.4 / 75.3 | 61.6 / 75.7 | 62.1 / 67.2 | 19.8 / 35.2 | 24.6 / 42.2 | 21.2 / 33.6 | 75.0 / 87.3 | 73.3 / 79.0 | 72.7 / 77.7 | 12.1 / 23.8 | 9.8 / 14.1 | 10.5 / 15.6 |
> > >
> > >
> > > In the above table, each cell reports both the final accuracy (Pass@1) and the ever-correct rate (EverPass@1 | t).  Across all configurations, we observe the same consistent pattern: the gap between Pass@1 and EverPass@1 | t persists, indicating that correct answers frequently appear at intermediate steps but are later overwritten during denoising.
> > >
> > > In the above table, we examine the effect of three orthogonal sources of randomness, including different random seeds temperature, and remasking strategies (low-confidence remasking vs. random remasking). More specifically:
> > >
> > > - Different seeds (42, 66, 218): Both metrics fluctuate slightly, but the Pass@1 vs. EverPass gap remains stable, demonstrating that temporal oscillation is not driven by initialization noise.
> > > - Different temperatures (0.0, 0.6, 1.0): Increasing temperature predictably affects absolute accuracy, yet the temporal oscillation gap consistently persists across all settings, showing that the effect is not merely a by-product of sampling randomness.
> > > - Different remasking strategies (low-confidence vs. random remasking): Random remasking substantially lowers overall accuracy, but even in this extreme setting, the EverPass@1|t curve remains substantially higher than Pass@1, confirming that oscillation persists even when the remasking policy is drastically perturbed.
> > >
> > > Taken together, **these results demonstrate that temporal oscillation is robust to decoding choices and cannot be eliminated by simply changing scheduling or inference parameters**. The consistency of the gap across all conditions strongly indicates that temporal oscillation is an inherent property of diffusion LLMs, rooted in the iterative denoising mechanism itself. This further motivates the need for methods such as temporal self-consistency voting and temporal consistency reinforcement.
> > >
> > > We have incorporated the corresponding experiments and discussions into Sec. C.5 of the revised manuscript.

---

> ### Author Response · Authors · 2025-11-20
> **Response to Reviewer 39aQ (Part 4)**
>
> # Q2: Alternative temporal consistency metrics or rewards
>
> Thank you for this helpful suggestion. Following the reviewer’s idea, we implemented the **pairwise agreement (PA)** metric at both the analysis level and as an RL reward. In the revision, we have added the full PA-based analysis and RL results to Sec. E.3 of the Appendix.
>
> **(1) Definition of Pairwise Agreement**
>
> Concretely, during decoding, we obtain a sequence of $T$ intermediate predictions  $\{｛\mathbf{x}_0^t\}｝ _{t=1}^T$. Based on this trajectory, we define the **change rate** as the fraction of consecutive steps that produce different answers:
>
> $$\operatorname{ChangeRate}=\frac{1}{T-1}\sum_{t=1}^{T-1} \mathbf{1} [\mathbf{x}_0^{t} = \mathbf{x}_0^{t+1}]$$
>
> We then define **pairwise agreement (PA)** as the complement of this change rate:
> $$
> \mathrm{PA} = 1 - \mathrm{ChangeRate}
> $$ so that higher PA corresponds to more consistent predictions across successive timesteps. We use PA as a trajectory-level temporal-consistency reward in our reinforcement learning experiments.
>
>
> **(2) Statistical Analysis**
>
> |          | GSM8K 128 | GSM8K 256 | GSM8K 512 | MATH500 128 | MATH500 256 | MATH500 512 | SVAMP 128 | SVAMP 256 | SVAMP 512 | Countdown 128 | Countdown 256 | Countdown 512 |
> |----------|-----------|-----------|-----------|-------------|-------------|-------------|-----------|-----------|-----------|---------------|---------------|---------------|
> | **Correct**  | 0.2206 | 0.1793 | 0.0249 | 0.3573 | 0.5162 | 0.5506 | 0.1391 | 0.0602 | 0.0151 | 0.6917 | 0.5778 | 0.2856 |
> | **Incorrect**| 0.2761 | 0.1833 | 0.0393 | 0.4535 | 0.5659 | 0.6349 | 0.2319 | 0.1148 | 0.0240 | 0.6992 | 0.5811 | 0.3755 |
>
> **First**, analogous to our original TSE analysis, we examined PA for finally correct vs. incorrect trajectories across GSM8K, MATH500, SVAMP, and Countdown with varing generation lengths (128/256/512).
> In all cases, finally incorrect trajectories exhibit a substantially *higher* change rate (and thus *lower* PA) than finally correct ones. This confirms that PA is indeed a strong temporal-consistency signal that can statistically distinguish correct from incorrect trajectories, consistent with the reviewer’s intuition.
>
> **(3) Reinforcement Learning Comparison**
>
> |                  | GSM8K 128 | GSM8K 256 | GSM8K 512 | MATH500 128 | MATH500 256 | MATH500 512 | SVAMP 128 | SVAMP 256 | SVAMP 512 | Countdown 128 | Countdown 256 | Countdown 512 |
> |------------------|-----------|-----------|-----------|-------------|-------------|-------------|-----------|-----------|-----------|---------------|---------------|---------------|
> | **Baseline**         | 70.2 | 78.7 | 80.1 | 23.8 | 34.4 | 36.8 | 81.7 | 83.3 | 82.7 | 21.5 | 19.9 | 21.5 |
> | **TSE**           | 72.2 | 78.8 | 80.2 | 30.6 | 34.6 | 38.0 | 84.3 | 89.0 | 88.7 | 38.6 | 53.5 | 44.9 |
> | **PA**      | 71.3 | 78.9 | 79.7 | 31.0 | 33.0 | 37.6 | 83.0 | 87.3 | 86.7 | 32.0 | 44.7 | 41.3 |
> | **TSE & Accuracy**   | 72.1 | 80.0 | 83.0 | 31.2 | 35.4 | 41.4 | 85.0 | 90.3 | 92.3 | 41.5 | 42.6 | 54.7 |
> | **PA & Accuracy** | 71.0 | 79.2 | 81.4 | 32.0 | 33.8 | 39.6 | 85.0 | 89.7 | 88.7 | 36.9 | 43.4 | 50.5 |
>
> **Second**, we used PA as the RL reward for Temporal Consistency Reinforcement and compared it with our TSE-based reward:
> - Using **PA alone** as the reward already improves over the supervised baseline on most benchmarks and sequence lengths.
> - **Combining PA with the accuracy reward** further boosts performance, mirroring the behavior of our "TSE + accuracy" setting.
> - Quantitatively, PA-based rewards achieve comparable gains to TSE-based rewards, while TSE is slightly better on average, especially in the more challenging settings (e.g., Countdown at longer sequence lengths), where trajectories are noisier and contain more diverse intermediate predictions.
>
> **Empirically**, the advantage of TSE over PA is therefore *modest but consistent*: both metrics are effective temporal-consistency measures, but TSE yields slightly larger and more robust improvements, particularly on harder tasks and longer sequences.
>
> Intuitively, **PA** captures *dynamic stability*—it can tell that "010101" is less stable than "000111"—but cannot differentiate between semantically diverse outputs such as "010203" vs. "010101." In contrast, **TSE** measures the *semantic dispersion* of intermediate predictions, capturing how the meaning of the partial answers drifts along the trajectory, but it is less sensitive to fine-grained dynamical fluctuations among semantically similar states.
>
> Looking forward, we believe an ideal temporal-consistency metric should combine the complementary strengths of PA and TSE—leveraging PA’s sensitivity to dynamics and TSE’s sensitivity to semantic variation. We plan to explore such hybrid rewards in future work.

---

> ### Author Response · Authors · 2025-11-20
> **Response to Reviewer 39aQ (Part 5)**
>
> # Q3: Minor typos/comments
>
> Thank you for the helpful suggestions. We have revised the manuscript accordingly and updated Sec. 3.1, Sec. 3.2, and Sec. 5.2, with all modifications highlighted in red. In particular, we corrected Eq. 1 to explicitly account for the marginalization over intermediate clean predictions. The revised formulation now reads:
> $$
> p _ {\theta}(\mathbf{x}_{0:T}) = p _ {\theta}(\mathbf{x}_T)\prod _ {t=1}^T p _ {\theta}(\mathbf{x} _ {t-1} \mid \mathbf{x} _ t)
> = p _ {\theta}(\mathbf{x} _ T)\prod _ {t=1}^T \big[\sum _ {\mathbf{x} _ 0^t} q(\mathbf{x} _ {t-1} \mid \mathbf{x} _ 0^t) p _ {\theta}(\mathbf{x} _ 0^t \mid \mathbf{x} _ t)\big].
> $$
> This revision resolves the inconsistency between the LHS and RHS and aligns the notation with the later definition of $\mathbf{x}_0^t$. We also corrected the typographical error in the caption ("Fig. 5b" → "Fig. 5a") and unified the notation throughout the manuscript to ensure clarity.
>
>
> In addition, we incorporated a focused discussion of concurrent studies examining temporal behaviors in diffusion language models and clarified how our approach complements these lines of work. These revisions appear in Sec. 2 of the updated manuscript.

---

> ### Author Response · Authors · 2025-11-27
> **Welcome Further Feedback**
>
> Dear Reviewer **39aQ**,
>
> We would like to thank you for the thoughtful and constructive feedback. During the rebuttal, we performed additional experiments and analyses and revised the manuscript accordingly to address your concerns. The experiments and analyses in our response are summarized as follows.
> - We expand empirical validation on new tasks.
>   - We add experiments on two natural-language QA benchmarks (ARC-C and Winogrande).
>   - We show that both Temporal Self-Consistency Voting and Temporal Consistency Reinforcement generalize beyond math reasoning tasks (Table S4, S5).
> - We explore alternatives to Temporal Semantic Entropy.
>   - We evaluate other temporal-consistency metrics, including pairwise agreement (PA) and token-level answer entropy.
>   - We analyze their behavior and discuss future directions for designing better temporal-dynamics metrics in dLLMs (Table S7; Fig. S5).
> - We analyze the causes of temporal oscillation.
>   - We study the effects of randomness, temperature, and different remasking strategies.
>   - We show that Pass@1 vs. EverPass@1|t gap persists universally, confirming temporal oscillation as an inherent property of dLLMs (Table S2).
>   - We provide a training-dynamics-based explanation (Sec. C.5).
> - We acknowledge that temporal voting relies on some baseline intermediate correctness, and clarify that combining TSE with accuracy rewards in RFT mitigates this reliance by providing a stabilizing complementary signal.
> - We improve writing quality and clarity.
>   - We enhance the mathematical rigor of formulas and fix minor typos.
>   - We strengthen the discussion of concurrent and related work.
>
> We are wondering if the responses have addressed your concerns, and would appreciate it if you considered raising your final rating. Looking forward to your further feedback!

---

### Official Review · Reviewer_B4qA · 2025-10-28

**Soundness:** 3
**Presentation:** 4
**Contribution:** 3
**Rating:** 4
**Confidence:** 4

**Summary:**

This paper reveals a critical phenomenon for diffusion language models: temporal oscillation. The authors observed that when answering a question, diffusion language models may generate a correct answer in the middle but replace it with an incorrect answer in the final output. The authors also conducted an empirical study to quantify this phenomenon. Based on this observation, the authors proposed two methods to enhance performance for diffusion language models: temporal self-consistency voting, a training-free majority voting method with minimal computational overhead; and temporal consistency reinforcement, a GRPO reinforcement learning algorithm that improves semantic consistency during post-training. For both methods, the authors observe improvements in accuracy across four reasoning benchmarks.

**Strengths:**

* This paper is well-written and easy to follow.
* This paper handles an interesting phenomenon that is distinct for diffusion language models. The authors conducted a quantitative analysis of temporal oscillations and proposed practical guidelines for leveraging them to improve the reasoning performance of diffusion language models.
* The experimental section of this paper is relatively abundant, with many ablations and extra analyses.

**Weaknesses:**

* The mathematical notation system is not rigorous and self-consistent. In equation 1, the authors tried to demonstrate the sampling algorithm of LLaDA-like diffusion language models. Although I understand what the authors were trying to convey, Eq. 1 is incorrect and confusing from a probabilistic perspective, and I strongly advise the authors to revise its presentation. Throughout the paper, the authors made no distinction between scalar values and tensors. For example, the $x_t$ in Eq. 1 represents a token sequence and is neither bolded nor indicated as a tensor. In Eq.3, it is not clear what is meant by $p(x)$, despite its importance. In Eq. 5, a minus sign is missing when it means a loss, but not when it means the policy gradient.
* In Figure 4, it was not indicated what the error bars meant. I do not understand why the error bar for the correct column for SVAMP dips below 0, since the entropy values should not be negative.
* In Table 1, the improvements of temporal majority voting seem marginal. I think more justifications, such as error bars, are needed to prove that these improvements are real but not random noise.

**Questions:**

* My biggest question is how the authors extract answers for intermediate token sequences. This is confusing because for LLaDA, once a token is decoded, it does not change throughout the sampling process, unless predictor-corrector samplers such as ReMDM are used. Thus, it is counterintuitive that the time oscillation has such a significant influence on the model's performance, as shown in Figure 1(a). Besides, it is not clear how to extract an answer from a partially masked token sequence such as "[MASK] [MASK] the [MASK]". Can the authors explain this in more detail, because it is clear that this is vital for both temporal self-consistency voting and the computation of TSE.
* Can the authors explain why TSE is chosen as the metric to use? The current explanation is Figure 4, but such a phenomenon holds true for many other metrics. For temporal consistency reinforcement, how will the performance be if TSE is replaced with token entropy? If they perform similarly, it is not clear why we should choose TSE. If they perform differently, can the authors provide some justification for why TSE is a good metric?
* What algorithms do the authors use for clustering when computing TSE? Is TSE robust to different clustering algorithms?
* Does temporal consistency reinforcement improve ever pass as well? What are the results if we apply temporal self-consistency voting to both the original checkpoint and the checkpoint after consistency reinforcement?
* In previous works such as d1, and wd1, Sudoku is also a commonly used benchmark. Can the authors provide results on the Sudoku benchmark to further justify the method?

---

> ### Author Response · Authors · 2025-11-20
> **Response to Reviewer B4qA (Part 1)**
>
> # W1: The mathematical notation system is not rigorous and self-consistent.
>
> Thank you for the detailed and constructive suggestions. We have thoroughly revised the mathematical notation system to ensure probabilistic rigor and internal consistency.
>
> **(1) Revision of Eq. 1**
> We agree that the original Eq. 1 lacked a valid probabilistic factorization. In the revised version, we now provide a probabilistically sound description of the sampling mechanism in LLaDA-style diffusion language models:
>
> $$
> p _ {\theta}(\mathbf{x}_{0:T}) = p _ {\theta}(\mathbf{x}_T)\prod _ {t=1}^T p _ {\theta}(\mathbf{x} _ {t-1} \mid \mathbf{x} _ t)
> = p _ {\theta}(\mathbf{x} _ T)\prod _ {t=1}^T \big[\sum _ {\mathbf{x} _ 0^t} q(\mathbf{x} _ {t-1} \mid \mathbf{x} _ 0^t) p _ {\theta}(\mathbf{x} _ 0^t \mid \mathbf{x} _ t)\big].
> $$
>
> Here, the summation over $\mathbf{x}_0^t$ runs over all possible clean intermediate predictions, making explicit the marginalization required for a valid probabilistic formulation.
>
> **(2) Clarification of Eq. 3**
> To eliminate ambiguity in the definition of cluster probability, we explicitly define:
> $$
> P(C_k)=\sum_{\mathbf{x}_0^t}\mathbf{1}[\mathbf{x}_0^t\in C_k]/T = |C_k|/T .
> $$
>
> Accordingly, the TSE becomes:
> $$
> \operatorname{TSE} = -\sum_{k=1}^K P(C_k)\log P(C_k).
> $$
>
> **(3) Correction of Eq. 5**
> We optimize the policy model with the following objective, which makes the sign convention explicit:
>
> $$
> \mathcal{J} _ {\mathrm{GRPO}}(\theta) = \mathbb{E}[\cdots] - \beta\, D_{\mathrm{KL}}(\pi _ \theta\|\pi _ {\mathrm{ref}}).
> $$
>
>
> **(4) Tensor vs. scalar notation**
> All tensor-valued variables (e.g., token sequences $\mathbf{x}_t$) are now consistently bolded using $\mathbf{x}$. We updated all notation in Sec. 3 and Sec. 4.
>
> Overall, these revisions substantially improve the correctness, clarity, and consistency of the mathematical presentation.

---

> ### Author Response · Authors · 2025-11-20
> **Response to Reviewer B4qA (Part 2)**
>
> # W2: In Fig. 4, it was not indicated what the error bars meant.
>
> Thank you for pointing this out. The error bars in Fig. 4 correspond to the mean ± standard deviation of TSE across all validation samples. Since TSE is mathematically non-negative, no individual sample ever attains a negative value. In the original plot, the lower end of the mean − std interval could visually extend below zero for categories with very small means (e.g., the "Finally-Correct" group on SVAMP). This graphical artifact arises solely from plotting the symmetric standard-deviation band; it does not indicate that any TSE values were negative.
>
> Following your suggestion, we have updated the figure to clip error bars at 0 so that the visualization respects the theoretical lower bound of TSE. We have also added revised caption as follows:
> "Error bars denote the mean ± standard deviation of TSE, clipped at 0 to respect its non-negative range."
>
> ---
>
> # W3: In Table 1, the improvements of temporal majority voting seem marginal. I think more justifications, such as error bars, are needed to prove that these improvements are real but not random noise.
>
> We appreciate the reviewer’s suggestion. To verify that the gains from temporal majority voting are not due to randomness, we compute **mean and standard deviation** from five repeated experiments. Table 1 has been updated accordingly.
>
>
> **LLaDA-8B-Instruct**
>
> | Model / Method | GSM8K 128 | GSM8K 256 | GSM8K 512 | MATH500 128 | MATH500 256 | MATH500 512 | SVAMP 128 | SVAMP 256 | SVAMP 512 | Count 128 | Count 256 | Count 512 |
> |---------------|---------|-----------|-----------|-----------|--------------|--------------|--------------|------------|-------------|------------|------------|------------|
> | LLaDA-8B-Instruct baseline | 68.3 ± 0.3 | 76.4 ± 0.3 | 78.1 ± 0.2 | 27.6 ± 0.4 | 33.2 ± 0.3 | 35.8 ± 0.4 | 84.3 ± 0.2 | 83.0 ± 0.3 | 84.3 ± 0.2 | 20.4 ± 0.4 | 21.4 ± 0.4 | 16.5 ± 0.5 |
> | + Fixed Weighting  | 68.0 ± 0.2 | 73.4 ± 0.2 | 78.3 ± 0.2 | 26.6 ± 0.2 | 30.8 ± 0.2 | 34.2 ± 0.2 | 87.0 ± 0.2 | 84.3 ± 0.2 | 84.3 ± 0.2 | 22.7 ± 0.2 | 18.8 ± 0.2 | 11.3 ± 0.2 |
> | + Linear Weighting  | 70.0 ± 0.2 | 78.0 ± 0.2 | 78.8 ± 0.2 | 28.0 ± 0.2 | 34.4 ± 0.2 | 34.6 ± 0.2 | 87.0 ± 0.2 | 84.3 ± 0.2 | 84.3 ± 0.2 | 24.2 ± 0.2 | 21.9 ± 0.2 | 16.0 ± 0.2 |
> | + Exp. Weighting | 70.1 ± 0.2 | 78.8 ± 0.2 | 78.9 ± 0.2 | 28.6 ± 0.2 | 35.2 ± 0.2 | 36.2 ± 0.2 | 86.3 ± 0.2 | 84.0 ± 0.2 | 84.7 ± 0.2 | 24.9 ± 0.2 | 23.3 ± 0.2 | 16.5 ± 0.2 |
> | Δ improvement | +1.8 | +2.4 | +0.8 | +1.0 | +2.0 | +0.4 | +2.0 | +1.0 | +0.4 | +4.5 | +1.9 | +0.0 |
> | EverPass@1 | 80.5 ± 0.3 | 85.2 ± 0.3 | 80.3 ± 0.3 | 40.2 ± 0.4 | 45.6 ± 0.4 | 47.2 ± 0.4 | 91.3 ± 0.2 | 89.3 ± 0.3 | 86.7 ± 0.3 | 28.1 ± 0.4 | 27.7 ± 0.4 | 21.1 ± 0.4 |
>
>
> **LLaDA-1.5**
>
> | Model / Method | GSM8K 128 | GSM8K 256 | GSM8K 512 | MATH500 128 | MATH500 256 | MATH500 512 | SVAMP 128 | SVAMP 256 | SVAMP 512 | Count 128 | Count 256 | Count 512 |
> |--------------|-----------|-----------|-----------|--------------|--------------|--------------|------------|-------------|-------------|------------|------------|------------|
> | LLaDA-1.5 baseline | 69.9 ± 0.3 | 79.2 ± 0.3 | 81.0 ± 0.2 | 29.2 ± 0.4 | 32.2 ± 0.3 | 35.6 ± 0.4 | 85.3 ± 0.2 | 86.3 ± 0.2 | 83.0 ± 0.3 | 21.6 ± 0.4 | 21.0 ± 0.4 | 20.8 ± 0.4 |
> | + Fixed Weighting | 68.8 ± 0.2 | 75.7 ± 0.2 | 80.3 ± 0.2 | 27.4 ± 0.2 | 30.8 ± 0.2 | 34.6 ± 0.2 | 87.3 ± 0.2 | 85.3 ± 0.2 | 84.0 ± 0.2 | 23.4 ± 0.2 | 22.3 ± 0.2 | 18.8 ± 0.2 |
> | + Linear Weighting | 71.0 ± 0.2 | 79.8 ± 0.2 | 81.0 ± 0.2 | 29.2 ± 0.2 | 32.8 ± 0.2 | 35.8 ± 0.2 | 86.0 ± 0.2 | 87.0 ± 0.2 | 84.0 ± 0.2 | 24.2 ± 0.2 | 23.4 ± 0.2 | 19.1 ± 0.2 |
> | + Exp. Weighting | 70.8 ± 0.2 | 79.6 ± 0.2 | 81.1 ± 0.2 | 29.2 ± 0.2 | 33.0 ± 0.2 | 36.4 ± 0.2 | 85.7 ± 0.2 | 87.7 ± 0.2 | 84.0 ± 0.2 | 26.3 ± 0.2 | 24.9 ± 0.2 | 21.2 ± 0.2 |
> | Δ improvement | +0.9 | +0.4 | +0.1 | +0.0 | +0.8 | +0.8 | +0.4 | +1.4 | +1.0 | +4.7 | +3.9 | +0.4 |
> | EverPass@1 | 81.5 ± 0.3 | 88.9 ± 0.3 | 83.6 ± 0.3 | 39.8 ± 0.4 | 47.4 ± 0.4 | 49.2 ± 0.4 | 90.7 ± 0.2 | 90.3 ± 0.2 | 86.0 ± 0.3 | 30.5 ± 0.4 | 27.0 ± 0.4 | 25.4 ± 0.4 |
>
>
> The results show that **temporal majority voting consistently improves performance across all runs, with low standard deviations**, indicating that the observed gains are stable and not attributable to random fluctuations.

---

> ### Author Response · Authors · 2025-11-20
> **Response to Reviewer B4qA (Part 3)**
>
> # Q1: My biggest question is how the authors extract answers for intermediate token sequences.
>
> We thank the reviewer for raising this important point.
>
> In LLaDA, a token becomes fixed only when it is **not** remasked at a given diffusion step. If a token is selected for remasking, its previously predicted logits are discarded, the position is reset to [MASK], and the token is re-predicted in later steps. Thus, a token may change over time until it eventually survives the remasking operation, at which point it remains stable in subsequent steps.
> Crucially, for our analysis, we do **not** extract answers from partially masked sequences. Instead, at every diffusion step, we use the **predicted token sequence before the remasking operation as the intermediate output**. This pre-remask prediction is fully decoded and provides a complete, unmasked sequence at every step, even though only a subset of tokens will be retained for the next step. Therefore, we never attempt to decode answers from strings such as "[MASK] [MASK] the [MASK]"; **we always extract answers from fully decoded, unmasked intermediate sequences**.
>
> This "decode-before-remask" property explains why temporal oscillation can still be substantial. Although tokens that survive remasking become fixed thereafter, many positions continue to be remasked and re-predicted across diffusion steps, producing a sequence of fully decoded intermediate predictions that exhibit meaningful variability over time.
>
> The answer extraction itself is straightforward: the final answer is obtained by simple syntactic parsing of the `<answer>...</answer>` span. If an intermediate prediction does not contain a valid answer span—for example, if the model outputs no `<answer>...</answer>` block at that step—we omit that step from temporal self-consistency voting or semantic clustering, and use only the steps where a valid answer can be extracted.

---

> ### Author Response · Authors · 2025-11-20
> **Response to Reviewer B4qA (Part 4)**
>
> # Q2: Alternatives to TSE for measuring temporal dynamics, such as token entropy?
>
> We thank the reviewer for the insightful question.
>
> **Why token entropy is not sufficient.**
> As shown in Fig. 3 and Appx. Fig. S1, average token entropy indeed correlates with correctness to some extent, indicating that local per-step confidence is informative. However, token entropy only reflects the uncertainty within a single diffusion step, and cannot capture whether the model maintains consistent semantics across steps. Temporal oscillation is inherently a trajectory-level property (i.e., the model changes its final answer multiple times), and this temporal stability cannot be inferred from a step-wise metric. In contrast, TSE explicitly measures cross-step semantic stability, making it well aligned with the nature of temporal consistency reinforcement.
>
> **Experimental comparison of alternative rewards.**
> To address the reviewer’s concern, we conducted ablations replacing TSE with two alternatives:
> 1. **negative average token entropy (ATE)**, and
> 2. **pairwise agreement (PA)** between consecutive diffusion steps (as suggested by Reviewer 39aQ).
>
> The results are summarized below:
>
> |                  | GSM8K 128 | GSM8K 256 | GSM8K 512 | MATH500 128 | MATH500 256 | MATH500 512 | SVAMP 128 | SVAMP 256 | SVAMP 512 | Countdown 128 | Countdown 256 | Countdown 512 |
> |------------------|-----------|-----------|-----------|-------------|-------------|-------------|-----------|-----------|-----------|---------------|---------------|---------------|
> | **Baseline**          | 70.2 | 78.7 | 80.1 | 23.8 | 34.4 | 36.8 | 81.7 | 83.3 | 82.7 | 21.5 | 19.9 | 21.5 |
> | **TSE**               | 72.2 | 78.8 | 80.2 | 30.6 | 34.6 | 38.0 | 84.3 | 89.0 | 88.7 | 38.6 | 53.5 | 44.9 |
> |  **ATE**              | 69.1 | 77.7 | 79.5 | 27.4 | 32.2 | 37.2 | 82.7 | 85.3 | 85.0 | 22.9 | 23.4 | 30.5 |
> | **PA**                | 71.3 | 78.9 | 79.7 | 31.0 | 33.0 | 37.6 | 83.0 | 87.3 | 86.7 | 32.0 | 44.7 | 41.3 |
> | **TSE & Accuracy**    | 72.1 | 80.0 | 83.0 | 31.2 | 35.4 | 41.4 | 85.0 | 90.3 | 92.3 | 41.5 | 42.6 | 54.7 |
> |  **ATE & Accuracy**   | 70.2 | 77.2 | 80.2 | 30.8 | 34.8 | 38.4 | 84.7 | 86.3 | 85.3 | 28.1 | 29.7 | 35.9 |
> | **PA & Accuracy**     | 71.0 | 79.2 | 81.4 | 32.0 | 33.8 | 39.6 | 85.0 | 89.7 | 88.7 | 36.9 | 43.4 | 50.5 |
>
> Our findings are:
> - Using negative token entropy as the reward yields moderate improvement, indicating that local confidence is useful but insufficient. Gains are consistently smaller than with TSE.
> - The pairwise agreement reward performs better than token entropy and comparable to TSE.
>
> Intuitively, **PA** captures *dynamic stability*—it can tell that "010101" is less stable than "000111"—but cannot differentiate between semantically diverse outputs such as "010203" vs. "010101." In contrast, **TSE** measures the *semantic dispersion* of intermediate predictions, capturing how the meaning of the partial answers drifts along the trajectory, but it is less sensitive to fine-grained dynamical fluctuations among semantically similar states.
>
> We believe an ideal temporal-consistency metric should integrate the complementary strengths of PA and TSE—leveraging PA’s sensitivity to dynamics and TSE’s sensitivity to semantic variation—and we plan to explore such hybrid rewards in future work.
>
> In the revision, we have added analysis on alternative temporal dynamic metrics to Sec. E.3 of the Appendix and discuss PA as a strong alternative temporal-consistency metric, while keeping TSE as our primary choice due to its slightly stronger empirical performance.
>
> ---
>
> # Q3: The clustering algorithms for computing TSE?
> Thank you for the question. To compute TSE, we use a simple and deterministic clustering procedure: all intermediate answers are extracted from the `<answer>...</answer>` span, and clusters are formed by **exact equivalence of the extracted answers**. This avoids the instability and overhead associated with embedding-based semantic clustering.
>
> More sophisticated clustering methods—such as those based on semantic embeddings [1,2], commonly used for hallucination detection—could in principle be applied. However, they introduce significantly higher computational and engineering complexity, while offering limited practical benefit in our setting, where the extracted answers are short, well-structured, and easily comparable. For this reason, we opt for the simplest reliable method for computing TSE.
>
> [1] *Semantic Uncertainty: Linguistic Invariances for Uncertainty Estimation in Natural Language Generation*
> [2] *Detecting Hallucinations in Large Language Models using Semantic Entropy*

---

> ### Author Response · Authors · 2025-11-20
> **Response to Reviewer B4qA (Part 5)**
>
> # Q4: Does temporal consistency reinforcement improve ever pass as well? What if we apply temporal self-consistency voting after consistency reinforcement?
>
> Thank you for the valuable suggestion. We have added the results for both the ever-pass performance after temporal consistency reinforcement and the subsequent temporal majority voting after reinforcement learning in Fig. 6b and Appendix Table S9(S5 in original paper before revise). To facilitate discussion, we first summarize the key experimental findings below:
>
>
> **Performance of the RFT model trained with the negative TSE reward alone**
>
> | Method / Seq Len | GSM8K-128 | GSM8K-256 | GSM8K-512 | MATH500-128 | MATH500-256 | MATH500-512 | SVAMP-128 | SVAMP-256 | SVAMP-512 | Countdown-128 | Countdown-256 | Countdown-512 |
> |------------------|-----------|-----------|-----------|-------------|-------------|-------------|-----------|-----------|-----------|----------------|----------------|----------------|
> | RFT with TSE | 72.2 | 78.8 | 80.2 | 30.6 | 34.6 | 38.0 | 84.3 | 89.0 | 88.7 | 38.6 | 53.5 | 44.9 |
> | + Exp. Weighting | 72.6 | 79.2 | 80.6 | 30.8 | 35.0 | 38.0 | 85.3 | 89.0 | 88.7 | 39.5 | 53.9 | 45.7 |
> | Δ | +0.4 | +0.4 | +0.4 | +0.2 | +0.4 | +0.0 | +1.0 | +0.0 | +0.0 | +0.9 | +0.4 | +0.8 |
> | EverPass | 80.5 | 81.8 | 81.3 | 33.4 | 37.4 | 40.0 | 88.7 | 90.7 | 89.7 | 44.1 | 59.8 | 55.5 |
>
> **Performance of the RFT model trained with the combined TSE and accuracy rewards**
>
> | Method / Seq Len | GSM8K-128 | GSM8K-256 | GSM8K-512 | MATH500-128 | MATH500-256 | MATH500-512 | SVAMP-128 | SVAMP-256 | SVAMP-512 | Countdown-128 | Countdown-256 | Countdown-512 |
> |------------------|-----------|-----------|-----------|-------------|-------------|-------------|-----------|-----------|-----------|----------------|----------------|----------------|
> | RFT with Combined | 72.1 | 80.0 | 83.0 | 31.2 | 35.4 | 41.4 | 85.0 | 90.3 | 92.3 | 41.5 | 42.6 | 54.7 |
> | + Exp. Weighting | 72.6 | 81.1 | 83.3 | 31.6 | 35.8 | 41.4 | 86.3 | 90.7 | 92.3 | 41.5 | 42.7 | 55.1 |
> | Δ | +0.5 | +1.1 | +0.3 | +0.4 | +0.4 | +0.0 | +1.3 | +0.4 | +0.0 | +0.0 | +0.1 | +0.4 |
> | EverPass | 84.0 | 91.4 | 87.9 | 43.6 | 49.2 | 52.0 | 90.7 | 91.7 | 92.7 | 54.3 | 68.0 | 70.3 |
>
>
> **(1) Does temporal consistency reinforcement improve EverPass?**
> Our experiments reveal that the effect on the EverPass rate depends strongly on the reward design used during reinforcement. When the model is trained solely with the negative TSE reward, EverPass rate decreases for some datasets, like MATH500 and SVAMP. We suspect this behavior stems from the nature of the negative TSE objective: it explicitly encourages the model to follow low-entropy, highly deterministic trajectories. Although this improves final-step accuracy, the reduced semantic entropy narrows the sampling distribution and suppresses exploration, which in turn decreases the likelihood that at least one trajectory solves the task. This phenomenon is consistent with prior observations in RL-based refinement of LLMs, where verifiable rewards often increase pass@1 but reduce pass@k by diminishing output diversity [3].
>
> In contrast, when TSE is combined with the accuracy reward, EverPass rate improves consistently across all tasks and sequence lengths. The accuracy reward compensates for excessive entropy minimization and preserves beneficial diversity in intermediate predictions. As a result, the combined objective achieves gains both in final-step correctness and in ever-pass performance. These findings validate that the interaction between temporal-consistency regularization and the reward structure is crucial, and that the combined reward yields the best trade-off between stability and exploration.
>
> **(2) Temporal Self-consistency Voting after reinforcement learning?**
> We further examined whether Temporal Self-consistency Voting (TSV) remains effective after the model has been trained with Temporal Consistency Reinforcement. To this end, we applied TSV to two RL-trained checkpoints—one optimized with the negative TSE reward and the other with the combined TSE + accuracy reward. While the gains are somewhat smaller than those obtained when applying voting before RFT (as in Table 1), both RL-trained models still show modest improvements. This suggests that the temporal signals exploited by TSV are preserved even after reinforcement learning, and that the method continues to extract meaningful information from the model’s intermediate predictions.
>
> Overall, these results demonstrate that Temporal Consistency Reinforcement and Temporal Self-consistency Voting are complementary: reinforcement strengthens trajectory-level consistency, while voting extracts additional signal from the remaining temporal variability. When used together, they consistently yield stronger performance than either technique alone.
>
> [3] *Does Reinforcement Learning Really Incentivize Reasoning Capacity in LLMs Beyond the Base Model?*

---

> ### Author Response · Authors · 2025-11-20
> **Response to Reviewer B4qA (Part 6)**
>
> # Q5: In previous works such as d1, and wd1, Sudoku is also a commonly used benchmark. Can the authors provide results on the Sudoku benchmark to further justify the method?
> We thank the reviewer for the comment. The Sudoku results are presented in Appendix D (Table S3, S1 in original paper before revise), and we summarize our findings as follows.
>
> **(1) Fundamental limitation on Sudoku due to extremely low intermediate correctness.**
> As discussed in Appendix D.2, Sudoku represents a setting where the model’s intermediate predictions are almost always incorrect—average correctness across all denoising steps is below 5%, as shown in Fig. S4. Since our temporal self-consistency voting relies on the presence of at least a small fraction of correct intermediate outputs, the near-absence of such signals makes reliable voting impossible. Consequently, both voting and TSE-only RFT struggle on this dataset, highlighting a limitation of temporal-consistency-based methods when the underlying model lacks minimal task competence.
>
> **(2) Combining TSE with the accuracy reward still brings benefits.**
> Interestingly, as shown in Table S1, while using the negative TSE reward alone leads to weaker performance, **combining TSE with the accuracy reward noticeably improves results** and even outperforms the accuracy-only baseline. We hypothesize that incorporating TSE introduces a more fine-grained and complementary reward structure beyond the binary correct/incorrect signal, which can stabilize learning even on difficult tasks like Sudoku.
>
> Overall, the Sudoku experiments serve as an informative stress test for our method: they reveal that negative TSE alone is insufficient when the model almost never produces correct intermediate states. However, they also highlight that integrating TSE with the accuracy reward remains consistently beneficial—even under such challenging conditions—likely because the combined signal offers both coarse-grained correctness guidance and fine-grained temporal stability cues.

---

> > ### Comment · Reviewer_B4qA · 2025-11-23
> > **My concerns are addressed.**
> >
> > I want to thank the authors for their rebuttal. All my concerns are addressed, and I will raise my score to 6.

---

> > > ### Author Response · Authors · 2025-11-24
> > >
> > > Thank you for your thoughtful feedback and for taking the time to review our work! We are glad that our response has resolved your concerns and greatly appreciate your insightful comments, which have significantly strengthened the paper.

---

### Official Review · Reviewer_TojT · 2025-11-01

**Soundness:** 2
**Presentation:** 3
**Contribution:** 3
**Rating:** 6
**Confidence:** 3

**Summary:**

This paper investigates temporal oscillation in diffusion language models—where correct intermediate answers are later overwritten—and proposes two methods, Temporal Self-Consistency Voting and Temporal Consistency Reinforcement, to leverage this phenomenon. The work is well-motivated and empirically strong, showing consistent improvements across benchmarks, with large gains on Countdown. However, methodological novelty is limited: TSCV mainly adapts prior self-consistency ideas, and the proposed TSE metric does not directly capture temporal dynamics but instead reflects semantic stability. The related work section could be more focused on studies addressing similar temporal behaviors. Overall, solid experiments and clear presentation justify a weak accept.

**Strengths:**

S1 This paper proposes leveraging the rich intermediate hidden states generated during the diffusion process—not only for inference-time aggregation but also for post-training reinforcement. This breaks away from the conventional paradigm that focuses solely on the final denoised output and opens a new avenue for understanding and enhancing dLLMs.
S2 The authors define Temporal Semantic Entropy to quantify semantic stability during the generation process and further incorporate it as a self-supervised reward signal in reinforcement learning. This method consistently improves performance across datasets, with especially notable results on the Countdown task, validating the effectiveness of temporal consistency signals in improving both stability and accuracy.
S3 The experimental results are strong and convincing. The proposed methods deliver clear and consistent gains across multiple reasoning benchmarks (GSM8K, MATH500, SVAMP, Countdown) and across different model scales (LLaDA-8B, LLaDA-1.5). In particular, the Temporal Consistency Reinforcement achieves up to 25 percentage-point improvement on the Countdown dataset, substantially outperforming prior diffusion-based baselines.
S4 The evaluation is comprehensive and well-controlled, including ablations over weighting schemes, reinforcement reward formulations, and entropy metrics. The improvements are not marginal but significant and stable across output lengths and datasets, demonstrating the general applicability of the proposed approach.

**Weaknesses:**

W1 The proposed Temporal Self-Consistency Voting essentially adapts the self-consistency idea from Self-Consistency Improves Chain-of-Thought Reasoning in Language Models (Wang et al., 2022) to diffusion models. This adaptation lacks substantial algorithmic innovation, and the observed performance gains are smaller than those achieved in autoregressive models—raising doubts about the method’s unique contribution within the diffusion framework.
W2 The proposed TSE metric is based on semantic clustering to measure semantic variation across denoising steps. However, this metric reflects the stability of latent semantic distributions rather than true temporal dynamics. Since the clustering process depends on embedding space similarity rather than the evolution of diffusion timesteps, TSE may not adequately represent temporal consistency.
W3 In Section 4.1, the paper attributes the efficiency and scalability of Temporal Self-Consistency Voting to its design. However, these “efficient” properties largely arise from the inherent parallel generation architecture of diffusion models rather than the proposed method itself. In other words, the ability to obtain multiple intermediate outputs within a single sampling trajectory is a natural feature of diffusion mechanisms, not a unique contribution of this approach.
W4 The Related Work section contains a substantial amount of content describing the general development of diffusion models and reinforcement learning variants, but these discussions are only tangentially related to the paper’s specific focus on temporal oscillation and consistency. This makes the literature review feel somewhat disconnected from the central contribution and could be tightened to highlight works directly addressing intermediate-step dynamics or temporal consistency.
W5 While the experimental results are quantitatively strong, the paper provides limited qualitative or diagnostic analysis of why temporal oscillation occurs and under what conditions correct intermediate answers are lost. Without a clearer causal explanation or visualization of representative trajectories, it remains difficult to interpret whether the proposed improvements stem from genuine temporal stabilization or simply from additional ensembling and regularization effects.

**Questions:**

Q1 The claimed “temporal oscillation” phenomenon requires more systematic experimental validation to support its research significance. Autoregressive models can also produce correct intermediate answers but incorrect final outputs, typically due to decoding uncertainty or local likelihood fluctuations. Similarly, oscillations observed in diffusion models may simply result from randomness or resampling noise during denoising. To justify the proposed methods, the authors should provide additional evidence showing that the phenomenon is structural and exploitable rather than a random artifact.
Q2 The strong performance of TSE-based reinforcement is primarily observed on the Countdown dataset, while its effects on GSM8K, MATH500, and SVAMP are comparable to those of TSCV. If TSE is only effective for specific tasks or distributions, its generalizability remains questionable. The authors are encouraged to extend their experiments to a broader range of datasets and task types to better demonstrate the method’s robustness and applicability.
Q3 The Related Work section could be revised to better connect with the paper’s central theme. In particular, the authors are encouraged to compare their study with more recent works that explicitly discuss similar temporal phenomena in diffusion models. For example, Diffusion Language Models Know the Answer Before Decoding also reports that correct answers often appear in intermediate denoising steps but are overwritten in later iterations—a phenomenon conceptually similar to the “temporal oscillation” described in this paper.

---

> ### Author Response · Authors · 2025-11-20
> **Response to Reviewer TojT (Part 1)**
>
> # W1: The proposed Temporal Self-Consistency Voting merely extends self-consistency (Wang et al., 2022) to diffusion models, offering limited algorithmic novelty and modest performance gains compared to autoregressive counterparts.
>
> Thank you for the insightful comment. We acknowledge that our voting mechanism resembles the idea of self-consistency (Wang et al., 2022) in form, but the key distinction lies in where the multiple candidate solutions come from and what new insight our work contributes.
>
> **(1) No extra sampling cost**
> Self-consistency requires multiple full sampling trajectories, which is expensive for both AR models and dLLMs. In contrast, our method uses the intermediate outputs naturally produced within a single diffusion trajectory, incurring no additional forward passes. This makes our approach significantly more efficient.
>
>
> To quantitatively validate this efficiency, we include a runtime comparison in Appendix Sec. C.4. As shown in the table below, the extra cost introduced by voting is **minimal across all datasets and sequence lengths**, typically within **0.03–0.3 seconds** compared to the non-voting baseline.
>
> **Runtime comparison (in seconds) for temporal self-consistency voting**
> | Model | Method / Seq Len | GSM8K-128 | GSM8K-256 | GSM8K-512 | MATH500-128 | MATH500-256 | MATH500-512 | SVAMP-128 | SVAMP-256 | SVAMP-512 | Count-128 | Count-256 | Count-512 |
> |-------|-------------------|-----------|-----------|-----------|--------------|--------------|--------------|------------|------------|------------|------------|------------|------------|
> | **LLaDA-8B-Instruct** | w/o voting (baseline) | 4.48 | 10.76 | 29.24 | 4.47 | 10.87 | 29.45 | 4.23 | 10.31 | 28.45 | 4.59 | 10.99 | 29.58 |
> |  | w/ voting | 4.53 | 10.86 | 29.76 | 4.54 | 10.91 | 29.61 | 4.29 | 10.41 | 28.59 | 4.64 | 11.12 | 29.81 |
>
>
> As the results show, the overhead of Temporal Self-Consistency Voting is **negligible**, confirming that the proposed method yields additional accuracy gains **without sacrificing efficiency**—a benefit that is unique to diffusion LMs and not available in autoregressive self-consistency.
>
> **(2) Our contribution is the insight, not the voting mechanism itself**
> Prior work on dLLMs treats intermediate predictions as noise and discards them entirely.
> Our analysis uncovers that these intermediate predictions often contain useful and sometimes correct semantic signals—a phenomenon previously unreported. Temporal Self-Consistency Voting is one simple instantiation of leveraging this insight, but the contribution is discovering and validating the value of temporal information, not the specific voting rule.
>
> **(3) Compatibility with existing self-consistency**
> Our approach is complementary rather than a replacement: it can be combined with classical self-consistency by applying temporal voting within each trajectory before performing cross-trajectory voting. Thus, our method enhances the utility of diffusion-specific temporal signals while remaining compatible with established techniques.
> In summary, while the voting form is reminiscent of self-consistency, our contribution is identifying and exploiting a previously overlooked source of information inherent to diffusion decoding, enabling efficient accuracy gains without additional sampling cost and with full compatibility with AR-style self-consistency.

---

> ### Author Response · Authors · 2025-11-20
> **Response to Reviewer TojT (Part 2)**
>
> # W2: TSE may not adequately represent temporal consistency.
>
>
> Thank you for this helpful suggestion. Following the reviewer’s idea, we introduced the **pairwise agreement (PA)** metric—an alternative measure that captures temporal dynamics more explicitly—and compared it to TSE at both the analysis level and within our reinforcement learning framework. In the revision, we have added the full PA-based analysis and RL results to Sec. E.3 of the Appendix.
>
>
> **(1) Definition of Pairwise Agreement**
>
> Concretely, during decoding, we obtain a sequence of $T$ intermediate predictions  $\{｛\mathbf{x}_0^t\}｝ _{t=1}^T$. Based on this trajectory, we define the **change rate** as the fraction of consecutive steps that produce different answers:
>
> $$\operatorname{ChangeRate}=\frac{1}{T-1}\sum_{t=1}^{T-1} \mathbf{1} [\mathbf{x}_0^{t} = \mathbf{x}_0^{t+1}]$$
>
> We then define **pairwise agreement (PA)** as the complement of this change rate:
> $$
> \mathrm{PA} = 1 - \mathrm{ChangeRate}
> $$ so that higher PA corresponds to more consistent predictions across successive timesteps. We use PA as a trajectory-level temporal-consistency reward in our reinforcement learning experiments.
>
>
> **(2) Statistical Analysis**
>
> |          | GSM8K 128 | GSM8K 256 | GSM8K 512 | MATH500 128 | MATH500 256 | MATH500 512 | SVAMP 128 | SVAMP 256 | SVAMP 512 | Countdown 128 | Countdown 256 | Countdown 512 |
> |----------|-----------|-----------|-----------|-------------|-------------|-------------|-----------|-----------|-----------|---------------|---------------|---------------|
> | **Correct**  | 0.2206 | 0.1793 | 0.0249 | 0.3573 | 0.5162 | 0.5506 | 0.1391 | 0.0602 | 0.0151 | 0.6917 | 0.5778 | 0.2856 |
> | **Incorrect**| 0.2761 | 0.1833 | 0.0393 | 0.4535 | 0.5659 | 0.6349 | 0.2319 | 0.1148 | 0.0240 | 0.6992 | 0.5811 | 0.3755 |
>
> **First**, analogous to our original TSE analysis, we examined PA for finally correct vs. incorrect trajectories across GSM8K, MATH500, SVAMP, and Countdown with varing generation lengths (128/256/512).
> In all cases, finally incorrect trajectories exhibit a substantially *higher* change rate (and thus *lower* PA) than finally correct ones. This confirms that PA is indeed a strong temporal-consistency signal that can statistically distinguish correct from incorrect trajectories, consistent with the reviewer’s intuition.
>
> **(3) Reinforcement Learning Comparison**
>
> |                  | GSM8K 128 | GSM8K 256 | GSM8K 512 | MATH500 128 | MATH500 256 | MATH500 512 | SVAMP 128 | SVAMP 256 | SVAMP 512 | Countdown 128 | Countdown 256 | Countdown 512 |
> |------------------|-----------|-----------|-----------|-------------|-------------|-------------|-----------|-----------|-----------|---------------|---------------|---------------|
> | **Baseline**         | 70.2 | 78.7 | 80.1 | 23.8 | 34.4 | 36.8 | 81.7 | 83.3 | 82.7 | 21.5 | 19.9 | 21.5 |
> | **TSE**           | 72.2 | 78.8 | 80.2 | 30.6 | 34.6 | 38.0 | 84.3 | 89.0 | 88.7 | 38.6 | 53.5 | 44.9 |
> | **PA**      | 71.3 | 78.9 | 79.7 | 31.0 | 33.0 | 37.6 | 83.0 | 87.3 | 86.7 | 32.0 | 44.7 | 41.3 |
> | **TSE & Accuracy**   | 72.1 | 80.0 | 83.0 | 31.2 | 35.4 | 41.4 | 85.0 | 90.3 | 92.3 | 41.5 | 42.6 | 54.7 |
> | **PA & Accuracy** | 71.0 | 79.2 | 81.4 | 32.0 | 33.8 | 39.6 | 85.0 | 89.7 | 88.7 | 36.9 | 43.4 | 50.5 |
>
> **Second**, we used PA as the RL reward for Temporal Consistency Reinforcement and compared it with our TSE-based reward:
> - Using **PA alone** as the reward already improves over the supervised baseline on most benchmarks and sequence lengths.
> - **Combining PA with the accuracy reward** further boosts performance, mirroring the behavior of our "TSE + accuracy" setting.
> - Quantitatively, PA-based rewards achieve comparable gains to TSE-based rewards, while TSE is slightly better on average, especially in the more challenging settings (e.g., Countdown at longer sequence lengths), where trajectories are noisier and contain more diverse intermediate predictions.
>
> **Empirically**, the advantage of TSE over PA is therefore *modest but consistent*: both metrics are effective temporal-consistency measures, but TSE yields slightly larger and more robust improvements, particularly on harder tasks and longer sequences.
>
> Intuitively, **PA** captures *dynamic stability*—it can tell that "010101" is less stable than "000111"—but cannot differentiate between semantically diverse outputs such as "010203" vs. "010101." In contrast, **TSE** measures the *semantic dispersion* of intermediate predictions, capturing how the meaning of the partial answers drifts along the trajectory, but it is less sensitive to fine-grained dynamical fluctuations among semantically similar states.
>
> Looking forward, we believe an ideal temporal-consistency metric should combine the complementary strengths of PA and TSE—leveraging PA’s sensitivity to dynamics and TSE’s sensitivity to semantic variation. We plan to explore such hybrid rewards in future work.

---

> > ### Author Response · Authors · 2025-11-20
> > **Response to Reviewer TojT (Part 3)**
> >
> > # W3: The ability to obtain multiple intermediate outputs within a single sampling trajectory is a natural feature of diffusion mechanisms, not a unique contribution of this approach.
> >
> >
> > Thank you for the valuable comment. We fully agree that obtaining multiple intermediate outputs within a single sampling trajectory is a natural property of diffusion models, and we do not claim this as our contribution.
> >
> > Our contribution is instead in recognizing the overlooked value of these intermediate predictions and showing that they contain meaningful—and often correct—reasoning signals that prior dLLM methods completely discard. Existing work relies solely on the final-step output and implicitly treats intermediate predictions as noise. Our analysis reveals a consistent temporal oscillation pattern, where correct answers frequently appear at intermediate steps but are overwritten later.
> >
> > Building on this observation, we propose simple and effective mechanisms to exploit these signals—such as Temporal Self-Consistency Voting—without requiring additional sampling trajectories. Thus, although generating intermediate outputs is inherent to diffusion models, realizing their usefulness and designing methods that leverage them efficiently is the contribution of our work.
> >
> > ---
> >
> > # W4 & Q3: The Related Work section should better connect to prior studies on intermediate-step dynamics and temporal consistency.
> >
> > Thank you for the suggestion. We have added a focused discussion of concurrent work on temporal behaviors in diffusion language models and clarified how our approach complements these studies. The manuscript has been updated accordingly, as shown below.
> >
> > Several concurrent work also investigates temporal behaviors in diffusion language models. Li et al. (2025a) [1] show that dLLMs often generate correct intermediate answers that are later overwritten, and propose stopping generation early when such signals appear. He et al. (2025) [2] further demonstrate that temporal cues along the denoising trajectory can be directly exploited to improve reasoning. Xie et al. (2025) [3] introduce a step-aware RL method that aligns denoising steps with hierarchical reasoning to avoid unstructured refinement.
> > While these studies highlight the value of early-step signals or step-specific structure, our work takes a complementary direction by developing both a lightweight test-time voting method and an unsupervised reinforcement signal grounded in trajectory-level temporal consistency.
> >
> > [1] *Diffusion Language Models Know the Answer Before Decoding*
> > [2] *MDPO: Overcoming the Training-Inference Divide of Masked Diffusion Language Model*
> > [3] *Step-Aware Policy Optimization for Reasoning in Diffusion Large Language Models*

---

> > > ### Author Response · Authors · 2025-11-20
> > > **Response to Reviewer TojT (Part 4)**
> > >
> > > # W5: Why temporal oscillation occurs and under what conditions correct intermediate answers are lost.
> > >
> > >
> > > Thank you for the insightful question.
> > >
> > > **1. Why temporal oscillation occurs?**
> > >
> > > Temporal oscillation naturally arises in Diffusion LLMs because the iterative denoising process repeatedly predicts remasked tokens. **Ideally, multiple predictions over the same partially masked sequence would all recover the same clean output. In practice, however, two key factors cause the model to produce different completions at different steps:**
> > >
> > > **(a) Ambiguity in supervision leads to inconsistent denoising targets.**
> > > Diffusion LLMs learn a mapping from partially masked sequences to clean sequences. Ideally, all partially masked variants of a sequence (e.g., `"<MASK>b<MASK><MASK>"`, `"<MASK>bc<MASK>"`) should map to the same output such as `"abcd"`. However, the training dataset often contains multiple sequences with overlapping visible regions. **For example, if both `seq1 = abcd` and `seq2 = abce` exist in the corpus, then the same masked input `"<MASK>bc<MASK>"` is supervised toward two *different* outputs.**
> > >
> > > This produces an effect similar to the ambiguity observed in image diffusion models [4, 5]—making the model’s prediction inherently unstable for certain masking patterns. Consequently, during inference, two adjacent diffusion steps may decode to different plausible completions (`"abcd"` at step t, `"abce"` at step t+1), manifesting as temporal oscillation.
> > >
> > > **(b) Underfitting under certain masking patterns can also induce deviations.**
> > > Even without conflicting examples, some masked contexts at test time—such as `"<MASK>bc<MASK>"`—may be rarely observed during training. The model may not have fully learned a uniquely correct continuation, causing it to output a near-miss sequence such as `"abcx"` simply due to insufficient signal. This also leads to oscillation between semantically close answers.
> > >
> > >
> > > **2. When correct intermediate answers are lost**
> > >
> > > Under semi-autoregressive decoding with low-confidence remasking, tokens in low-confidence regions—including those in the answer span—may be remasked again **even after a correct prediction has already appeared**. Once these tokens are remasked, subsequent denoising steps can shift the trajectory toward another plausible completion, thereby overwriting the earlier correct answer.
> > >
> > > Our empirical findings strongly support this explanation. As shown in **Fig. 2** and **Appendix Fig. S1**, the ever-pass rate remains consistently higher than the final pass rate across both **harder datasets** (e.g., Countdown) and **simpler datasets** (e.g., GSM8K). In all cases, **temporal oscillation begins as soon as correct intermediate answers start to emerge**—indicating that correct predictions, once formed, are not stably preserved by the iterative denoising process. This underscores that overwriting correct states is a pervasive and dataset-agnostic behavior of current diffusion LLM decoding.
> > >
> > > We have incorporated the corresponding experiments and discussions into Sec. C.5 of the revised manuscript.
> > >
> > >
> > > [4] *Flow Matching for Generative Modeling*
> > > [5] *Flow Straight and Fast: Learning to Generate and Transfer Data with Rectified Flow*

---

> > > > ### Author Response · Authors · 2025-11-20
> > > > **Response to Reviewer TojT (Part 5)**
> > > >
> > > > # Q1: The claimed "temporal oscillation" phenomenon requires more systematic experimental validation to support its research significance.
> > > >
> > > >
> > > > Thank you for raising this important point. In the revised manuscript, we provide systematic and controlled experiments to validate that temporal oscillation is a robust and model-intrinsic phenomenon, rather than an artifact of a particular decoding configuration.
> > > >
> > > > | Seed | Temp. | Remasking | GSM8K 128 | GSM8K 256 | GSM8K 512 | MATH 128 | MATH 256 | MATH 512 | SVAMP 128 | SVAMP 256 | SVAMP 512 | Count. 128 | Count. 256 | Count. 512 |
> > > > |------|--------|-----------|------------|------------|------------|-----------|-----------|-----------|-------------|-------------|-------------|--------------|--------------|--------------|
> > > > | 42 | 0.0 | LowConf | 67.9 / 80.3 | 76.5 / 84.8 | 79.2 / 81.0 | 27.4 / 40.2 | 33.4 / 45.6 | 35.8 / 47.2 | 83.3 / 90.7 | 84.0 / 89.7 | 84.7 / 87.3 | 18.8 / 25.8 | 20.3 / 27.7 | 15.6 / 19.9 |
> > > > | 218 | 0.0 | LowConf | 68.2 / 81.5 | 76.5 / 84.8 | 79.2 / 81.0 | 27.2 / 40.8 | 33.4 / 45.6 | 35.8 / 47.2 | 83.0 / 91.0 | 84.0 / 89.7 | 84.7 / 87.3 | 20.3 / 29.3 | 20.3 / 27.0 | 16.0 / 20.7 |
> > > > | 66 | 0.0 | LowConf | 69.6 / 80.3 | 76.3 / 85.2 | 78.2 / 80.3 | 27.2 / 40.8 | 33.4 / 45.6 | 35.8 / 47.2 | 83.3 / 90.7 | 83.3 / 89.3 | 84.7 / 86.7 | 20.3 / 27.7 | 22.3 / 29.3 | 18.8 / 22.7 |
> > > > | 42 | 0.6 | LowConf | 67.4 / 82.6 | 76.6 / 86.7 | 78.5 / 81.5 | 25.6 / 43.2 | 33.8 / 49.0 | 33.8 / 50.8 | 84.7 / 91.7 | 85.0 / 90.7 | 84.7 / 87.0 | 20.3 / 29.3 | 22.3 / 20.7 | 18.8 / 22.7 |
> > > > | 42 | 1.0 | LowConf | 66.1 / 78.8 | 75.2 / 81.9 | 77.0 / 79.2 | 23.8 / 37.6 | 30.6 / 42.0 | 30.8 / 42.2 | 85.0 / 89.7 | 84.3 / 87.0 | 85.0 / 87.3 | 19.5 / 22.3 | 17.6 / 19.5 | 16.8 / 21.1 |
> > > > | 42 | 0.0 | Random | 57.2 / 76.9 | 64.5 / 76.6 | 62.1 / 66.8 | 19.8 / 35.8 | 24.4 / 38.6 | 23.4 / 33.4 | 73.0 / 86.0 | 73.3 / 82.0 | 72.7 / 78.0 | 10.9 / 22.3 | 8.2 / 12.5 | 10.5 / 14.5 |
> > > > | 218 | 0.0 | Random | 59.4 / 75.3 | 61.6 / 75.7 | 62.1 / 67.2 | 19.8 / 35.2 | 24.6 / 42.2 | 21.2 / 33.6 | 75.0 / 87.3 | 73.3 / 79.0 | 72.7 / 77.7 | 12.1 / 23.8 | 9.8 / 14.1 | 10.5 / 15.6 |
> > > >
> > > >
> > > > In the above table, each cell reports both the final accuracy (Pass@1) and the ever-correct rate (EverPass@1 | t).  Across all configurations, we observe the same consistent pattern: the gap between Pass@1 and EverPass@1 | t persists, indicating that correct answers frequently appear at intermediate steps but are later overwritten during denoising.
> > > >
> > > > In the above table, we examine the effect of three orthogonal sources of randomness, including different random seeds temperature, and remasking strategies (low-confidence remasking vs. random remasking). More specifically:
> > > >
> > > > - Different seeds (42, 66, 218): Both metrics fluctuate slightly, but the Pass@1 vs. EverPass gap remains stable, demonstrating that temporal oscillation is not driven by initialization noise.
> > > > - Different temperatures (0.0, 0.6, 1.0): Increasing temperature predictably affects absolute accuracy, yet the temporal oscillation gap consistently persists across all settings, showing that the effect is not merely a by-product of sampling randomness.
> > > > - Different remasking strategies (low-confidence vs. random remasking): Random remasking substantially lowers overall accuracy, but even in this extreme setting, the EverPass@1|t curve remains substantially higher than Pass@1, confirming that oscillation persists even when the remasking policy is drastically perturbed.
> > > >
> > > > Together, these results demonstrate that temporal oscillation is robust across decoding randomness, sampling noise, and remasking behavior. These findings strongly support that temporal oscillation is an inherent property of diffusion LLMs, rather than an artifact of specific choices in decoding configuration.
> > > >
> > > > We have incorporated the corresponding experiments and discussions into Sec. C.5 of the revised manuscript.

---

> > > > > ### Author Response · Authors · 2025-11-20
> > > > > **Response to Reviewer TojT (Part 6)**
> > > > >
> > > > > # Q2: The authors are encouraged to extend their experiments to a broader range of datasets and task types to better demonstrate the method’s robustness and applicability.
> > > > >
> > > > > Thank you for the valuable suggestion. To verify that our methods generalize beyond math reasoning, we added experiments on two natural language QA benchmarks, **ARC-C** and **Winogrande**, evaluating both **Temporal Self-Consistency Voting** and **Temporal Consistency Reinforcement**.
> > > > >
> > > > > **(1)  Temporal Self-Consistency Voting**
> > > > > We follow exactly the same **semi-autoregressive** decoding setup as in the main paper (block size 32).  For exponential weighting we reuse the hyperparameter **α = 1.5** tuned on math datasets, without any tuning on QA. The main voting results is shown below:
> > > > >
> > > > > | Method / Seq Len | ARC-C-128 | ARC-C-256 | ARC-C-512 | Winogrande-128 | Winogrande-256 | Winogrande-512 |
> > > > > |------------------|------|------|------|------|------|------|
> > > > > | **Baseline**     | 81.4 | 83.3 | 83.1 | 53.1 | 53.6 | 52.5 |
> > > > > | **Fixed Weighting** | 82.6 | 83.8 | 83.4 | 54.4 | 55.1 | 55.0 |
> > > > > | **Linear Weighting** | 82.6 | 83.8 | 83.4 | 56.1 | 57.7 | 58.4 |
> > > > > | **Exp. Weighting (α=1.5)** | 82.6 | 83.8 | 83.4 | 54.4 | 55.1 | 55.0 |
> > > > > | **Δ (Exp − Base)** | +1.2 | +0.5 | +0.3 | +1.3 | +1.5 | +2.5 |
> > > > > | **EverPass** | 86.1 | 86.9 | 85.5 | 61.5 | 61.6 | 62.0 |
> > > > >
> > > > > On **ARC-C**, exponential temporal voting improves accuracy by about **+0.7 points on average** over the baseline. On **Winogrande**, the untuned exponential weighting yields about **+1.8 points on average**, and **linear weighting** further improves performance, reaching up to **58.4** (vs. **52.5** baseline at length 512). These results indicate that temporal voting yields consistent gains on natural language QA and is robust to the choice of weighting scheme.
> > > > >
> > > > > **(2)  Temporal Consistency Reinforcement**
> > > > > For **Temporal Consistency Reinforcement**, we train on ARC-C and Winogrande with the same GRPO setup as in the main experiments, using either (i) a **negative TSE reward** alone or (ii) a **combined TSE + accuracy reward**. Both configurations consistently outperform the baseline across all sequence lengths on ARC-C, yielding gains of about 2–3 points (e.g., from 81.6→84.5 at 128 tokens and 82.8→85.8 at 256 tokens), while on Winogrande the combined reward provides the largest improvements, with up to +12.4 points (e.g., 56.6→69.0 at 256 tokens). The numbers are detailed in the following table.
> > > > >
> > > > >
> > > > > | Method / Seq Len          | ARC-C-128 | ARC-C-256 | ARC-C-512 | Winogrande-128 | Winogrande-256 | Winogrande-512 |
> > > > > |---------------------------|-----------|-----------|-----------|----------------|----------------|----------------|
> > > > > | **Baseline**              | 81.6      | 82.8      | 83.4      | 54.5           | 56.6           | 56.5           |
> > > > > | **TSE only**              | 84.5      | 84.8      | 84.8      | 59.1           | 60.6           | 61.3           |
> > > > > | **Δ (TSE − Base)**        | +2.9      | +2.0      | +1.4      | +4.6           | +4.0           | +4.8           |
> > > > > | **Combined**              | 84.5      | 85.8      | 86.3      | 66.6           | 69.0           | 68.5           |
> > > > > | **Δ (Combined − Base)**   | +2.9      | +3.0      | +2.9      | +12.1          | +12.4          | +12.0          |
> > > > >
> > > > >
> > > > > Overall, the additional experiments on **ARC-C** and **Winogrande** demonstrate that both our test-time voting and reinforcement learning methods **generalize beyond math reasoning to natural language QA**, supporting our claim that exploiting temporal dynamics is a broadly applicable principle for improving diffusion language models. We have incorporated these results into **Sec. E.1** of the revised manuscript.

---

> ### Author Response · Authors · 2025-11-27
> **Welcome Further Feedback**
>
> Dear Reviewer **TojT**,
>
> We would like to thank you for the thoughtful and constructive feedback. During the rebuttal, we performed additional experiments and analyses and revised the manuscript accordingly to address your concerns. The experiments and analyses in our response are summarized as follows.
> - We explore alternatives to Temporal Semantic Entropy.
>   - We evaluate other temporal-consistency metrics, including pairwise agreement (PA) and token-level answer entropy.
>   - We analyze their behavior and discuss future directions for designing better temporal-dynamics metrics in dLLMs (Table S7; Fig. S5).
> - We analyze the causes of temporal oscillation.
>   - We study the effects of randomness, temperature, and different remasking strategies.
>   - We show that Pass@1 vs. EverPass@1|t gap persists universally, confirming temporal oscillation as an inherent property of dLLMs (Table S2).
>   - We provide a training-dynamics-based explanation (Sec. C.5).
> - We expand empirical validation on new tasks.
>   - We add experiments on two natural-language QA benchmarks (ARC-C and Winogrande).
>   - We show that both Temporal Self-Consistency Voting and Temporal Consistency Reinforcement generalize beyond math reasoning tasks (Table S4, S5).
> - We clarify that our novelty lies in uncovering and leveraging informative intermediate diffusion predictions at no extra cost, providing efficient accuracy gains while remaining compatible with classical self-consistency.
> - We added a time-cost analysis showing that voting introduces only ~1% runtime overhead in Table S1.
> - We strengthen the discussion of concurrent and related work.
>
> We are wondering if the responses have addressed your concerns, and would appreciate it if you considered raising your final rating. Looking forward to your further feedback!

---

### Official Review · Reviewer_CBgE · 2025-11-01

**Soundness:** 3
**Presentation:** 3
**Contribution:** 3
**Rating:** 6
**Confidence:** 5

**Summary:**

This paper investigates temporal dynamics in diffusion language models (dLLMs), highlighting that correct intermediate predictions are often lost during later denoising steps. The authors identify a “temporal oscillation” phenomenon and propose two complementary methods: (1) Temporal Self-Consistency Voting, a test-time decoding strategy that aggregates predictions across steps, and (2) Temporal Consistency Reinforcement, a post-training method using Temporal Semantic Entropy (TSE) as a reward to encourage stable generations. Experiments show consistent improvements across reasoning benchmarks, suggesting that temporal signals within diffusion decoding are a valuable yet underexplored source of information.

**Strengths:**

1. The paper introduces a new metric, Temporal Semantic Entropy (TSE), to quantify semantic fluctuation across denoising steps, revealing meaningful patterns in dLLMs.

2. Proposes both training-free (voting-based decoding) and training-based (RL with TSE reward) methods, offering complementary ways to leverage temporal consistency.

3. Demonstrates that TSE can serve as a soft reward signal for RL even in unlabeled scenarios, expanding applicability to broader settings.

**Weaknesses:**

1. In Table 2, combining accuracy reward and TSE sometimes leads to negative effects under RFT, indicating potential instability in multi-reward optimization.

2. After RFT, the model’s everPass@1 performance decreases for some tasks (MATH500 and SVAMP), as shown in Table 1 and S4.

3. Using TSE as an RL reward requires semantic clustering, which introduces extra computation overhead not discussed in the paper.

4. The paper lacks clarity about which decoding strategy is used in the analysis experiments (e.g., random remarking, low-confidence remasking, or block decoding). It appears analysis assumes full diffusion (len=128, steps=64), whereas RL experiments adopt semi-autoregressive decoding—raising concerns about experimental consistency.

**Questions:**

1. In Figure 3, is the average token-level entropy computed over the last-step tokens only or all tokens across trajectories?

2. In Figure 4, why not separately report metrics for "finally correct" versus "intermediate correct" cases to better illustrate temporal oscillation behavior?

---

> ### Author Response · Authors · 2025-11-20
> **Response to Reviewer CBgE (Part 1)**
>
> # W1: Combining accuracy reward and TSE sometimes leads to negative effects under RFT
>
> Thank you for the insightful comment. Considering that dLLMs are known to be sensitive to generation length [1], we conducted an additional analysis focusing on the **best performance across all evaluated lengths**. This removes the confounding effect introduced by length-dependent behavior and isolates the impact of reward combination. The comparison of best accuracy across lengths between (i) negative TSE reward alone and (ii) combining accuracy reward with negative TSE reward is shown in the following table:
>
> | Model Variant | GSM8K | MATH500 | SVAMP | Countdown |
> |--------------|-------|---------|-------|-----------|
> | **LLaDA** | 80.1 | 36.8 | 83.3 | 21.5 |
> | + negative TSE reward | 80.2 | 38.0 | 89.0 | 53.5 |
> | + combining | **83.0** | **41.4** | **92.3** | **54.7** |
> | **LLaDA-1.5** | 81.1 | 35.4 | 86.3 | 21.5 |
> | + negative TSE reward | 82.6 | 40.0 | 88.3 | 55.9 |
> | + combining | **84.0** | **41.4** | **90.3** | **63.3** |
>
> Across all benchmarks (**GSM8K, MATH500, SVAMP, Countdown**) and both model variants (**LLaDA and LLaDA-1.5**), the **combined reward consistently achieves higher best-case accuracy** than using the negative-TSE reward alone. This demonstrates that, once controlling the choice of generation length, the combined reward provides more stable and more substantial improvements.
>
> We suspect the occasional declines observed in Table 2 of the main paper stem from the **intrinsic length sensitivity** of dLLMs. As shown in prior work [1], appropriate generation length or adaptive-length inference strategies can significantly influence the final answer quality. Under suboptimal length settings, the gains from RFT may not be fully reflected. Further exploration of **length-adaptive reward optimization** may help mitigate the length sensitivity of dLLMs and improve robustness.
>
> Beyond final pass rate, we examined the ever pass rate, which reflects the model’s ability to reach a correct answer at any timestep. We observe that although the negative-TSE reward alone sometimes yields competitive final accuracy, **the combined reward consistently achieves higher ever pass rate across all datasets and lengths, as shown below**:
>
> | Model / Seq Len           | GSM8K-128 | GSM8K-256 | GSM8K-512 | MATH500-128 | MATH500-256 | MATH500-512 | SVAMP-128 | SVAMP-256 | SVAMP-512 | Count-128 | Count-256 | Count-512 |
> |---------------------------|-----------|-----------|-----------|-------------|-------------|-------------|-----------|-----------|-----------|-----------|-----------|-----------|
> | **LLaDA**                 | 81.5      | 88.9      | 83.6      | 39.8        | 47.4        | 49.2        | **90.7**  | 90.3      | 86.0      | 30.5      | 27.0      | 25.4      |
> | **Negative TSE reward**   | 80.5      | 81.8      | 81.3      | 33.4        | 37.4        | 40.0        | 88.7      | 90.7      | 89.7      | 44.1      | 59.8      | 55.5      |
> | **TSE & accuracy reward** | **84.0**  | **91.4**  | **87.9**  | **43.6**    | **49.2**    | **52.0**    | **90.7**  | **91.7**  | **92.7**  | **54.3**  | **68.0**  | **70.3**  |
>
> The combined reward therefore not only improves final accuracy but also increases the model’s coverage of correct reasoning trajectories—a desirable property for diffusion-based decoding, where correct answers often appear at intermediate steps.
>
> [1] *Beyond Fixed: Training-Free Variable-Length Denoising for Diffusion Large Language Models*
> [2] *Large Language Diffusion Models*

---

> > ### Author Response · Authors · 2025-11-20
> > **Response to Reviewer CBgE (Part 2)**
> >
> > # W2: After RFT, the model’s everPass@1 performance decreases for some tasks (MATH500 and SVAMP)
> >
> > Thank you for the helpful observation. As shown in Table 1 and Appendix Table S6(Table S4 in original paper before revise), we report the baseline EverPass@1 results and the corresponding values after RFT with either the negative TSE reward or the combined TSE & accuracy reward. We summarize the relevant numbers below for clarity:
> >
> > | Model / Seq Len           | GSM8K-128 | GSM8K-256 | GSM8K-512 | MATH500-128 | MATH500-256 | MATH500-512 | SVAMP-128 | SVAMP-256 | SVAMP-512 | Count-128 | Count-256 | Count-512 |
> > |---------------------------|-----------|-----------|-----------|-------------|-------------|-------------|-----------|-----------|-----------|-----------|-----------|-----------|
> > | **LLaDA**                 | 80.5      | 85.2      | 80.3      | 40.2        | 45.6        | 47.2        | **91.3**  | 89.3      | 86.7      | 28.1      | 27.7      | 21.1      |
> > | **Negative TSE reward**   | 80.5      | 81.8      | 81.3      | 33.4        | 37.4        | 40.0        | 88.7      | 90.7      | 89.7      | 44.1      | 59.8      | 55.5      |
> > | **TSE & accuracy reward** | **84.0**  | **91.4**  | **87.9**  | **43.6**    | **49.2**    | **52.0**    | 90.7  | **91.7**  | **92.7**  | **54.3**  | **68.0**  | **70.3**  |
> >
> > We indeed observe that RFT using the negative TSE reward alone leads to a drop in EverPass@1 on MATH500 and SVAMP. Our analysis suggests that this effect is linked to the nature of the negative TSE reward: it explicitly encourages the model to follow low-semantic-entropy (more deterministic) trajectories. While this improves final-step stability and accuracy, it can reduce exploration during sampling, resulting in a narrower trajectory distribution and therefore a modest reduction in ever-pass rate.
> >
> > This behavior is consistent with prior findings on RL-based RFT in LLMs (e.g., [3]), which also report that after RL training, models tend to become more deterministic—improving pass@1 but sometimes reducing pass@k due to decreased output diversity. Thus, the phenomenon appears to be a general property of RL with verifiable rewards, not specific to our temporal-consistency objective.
> >
> > Importantly, when TSE is combined with the accuracy reward, this issue is mitigated. As shown in the table above, the combined reward consistently improves EverPass@1 across all tasks. We believe this is because the accuracy reward provides an additional supervision signal that counteracts excessive entropy reduction, preserving beneficial diversity while still encouraging temporal consistency.
> >
> > Overall, we view this behavior as an inherent trade-off in reward-driven RFT: negative TSE alone pushes the model toward stronger determinism, while the combined reward balances consistency and exploration, delivering improvements both in final-step accuracy and in ever-pass rate.
> >
> > [3] *Does Reinforcement Learning Really Incentivize Reasoning Capacity in LLMs Beyond the Base Model?*
> >
> > ---
> >
> > # W3: Using TSE as an RL reward requires semantic clustering, which introduces extra computation overhead
> >
> > Thank you for the comment. In practice, computing TSE does not introduce any additional semantic parsing or model inference beyond what accuracy-based rewards already require. We derive semantic clusters directly from the extracted final answer, without using any external clustering model.
> > Concretely, we obtain the final answer via simple syntactic parsing of the `<answer>...</answer>` span and group intermediate answers by exact numerical equivalence (i.e., identical extracted numbers define a cluster). This procedure mirrors the same answer-extraction step needed for computing accuracy rewards, which also depends on parsing a single numeric output from the generated text. Consequently, the computational overhead of TSE is negligible and on par with standard answer-extraction operations in existing RFT pipelines.
> > A comprehensive time-resolved analysis, along with the full numerical results, has been included in Appendix C.4 in the revised manuscript.

---

> > > ### Author Response · Authors · 2025-11-20
> > > **Response to Reviewer CBgE (Part 3)**
> > >
> > > # W4: The paper lacks clarity about which decoding strategy is used in the analysis experiments
> > > Thank you for pointing this out. We apologize for the insufficient clarity and have revised Sec. 3.3 and Sec. 5.1 to explicitly describe all decoding configurations used in our experiments. Below, we summarize the unified decoding setup:
> > >
> > > (1) Decoding strategy:
> > > - Across all experiments—including RL training rollouts, evaluation, and all analysis experiments—we adopt semi-autoregressive decoding with low-confidence remasking, and a block size of 32, following LLaDA [2] and d1 [4].
> > >
> > > (2) Generation lengths:
> > > - Rollouts in RL training: During RFT rollouts, we generate sequences of length 256, using 128 diffusion steps, following d1 [4].
> > > - Evaluation: For test-time self-consistency voting and all RL evaluations, we evaluate three generation lengths: 128, 256, 512 tokens corresponding to 64, 128, 256 diffusion steps, respectively (again following d1 [4]).
> > > - For the exploratory experiment in Sec. 3.2 / Fig. 1, we analyze generation lengths 128, 256, 512, and observe temporal oscillation under all settings. For the analyses in Sec. 3.3, we use the default 128-length / 64-step configuration. Appendix Sec. C.2 / Fig. S2 further reports the temporal semantic entropy analyses under generation lengths of 256 and 512 to demonstrate robustness.
> > >
> > > We have incorporated these clarifications into Sec. 5.1 and Sec. B.3 of the revised manuscript.
> > >
> > > ---
> > >
> > >
> > > # Q1: In Fig. 3, is the average token-level entropy computed over the last-step tokens only or all tokens across trajectories?
> > > Thank you for the question. In Fig. 3, our goal is to measure the model’s **average confidence over all tokens at each diffusion step**. Therefore, the average token-level entropy is computed using **all tokens predicted at that specific step**.
> > >
> > > Concretely, for a sequence of length $L$ at decoding step $t$, we take the logits of **all $L$ tokens predicted at step $t$** and compute their token-wise entropies, which are then averaged. This reflects the model’s overall uncertainty at each diffusion step and characterizes how confidence evolves throughout the denoising trajectory.
> > >
> > > ---
> > >
> > > # Q2: In Fig. 4, why not separately report metrics for "finally correct" versus "intermediate correct" cases to better illustrate temporal oscillation behavior?
> > >
> > > Thank you for the insightful suggestion. Following your feedback, we now explicitly separate the "finally correct" and "intermediate correct" cases in  Fig. 4 of the main paper. This comparison reveals a clear distinction: **trajectories that ultimately reach the correct answer consistently exhibit lower Temporal Semantic Entropy**, reflecting more stable and coherent temporal behavior. In contrast, **intermediate-correct trajectories show higher TSE in most cases**, indicating stronger semantic fluctuation and temporal oscillation. These results demonstrate that **TSE effectively differentiates "finally correct" and "intermediate correct" cases**, providing a clearer characterization of temporal dynamics during decoding. We have updated the corresponding sections of the main paper accordingly.

---

> ### Author Response · Authors · 2025-11-27
> **Welcome Further Feedback**
>
> Dear Reviewer **CBgE**,
>
> We would like to thank you for the thoughtful and constructive feedback. During the rebuttal, we performed additional experiments and analyses and revised the manuscript accordingly to address your concerns. The experiments and analyses in our response are summarized as follows.
> - Our additional analyses show that combining the accuracy reward with the negative-TSE reward consistently improves both best-case accuracy and ever-pass rate across datasets and model variants, confirming its more stable and substantial benefits. than using negative TSE alone.
> - While negative TSE alone can overly reduce trajectory diversity and lower ever-pass rates on some tasks, the combined TSE + accuracy reward restores this balance and consistently improves the ever-pass rate across all settings.
> - TSE incurs virtually no extra computation because its clustering relies only on the same simple answer-extraction step already required for standard accuracy rewards. Our time-cost analysis shows that voting introduces only ~1% runtime overhead in Table S1.
> - We have incorporated detailed decoding-strategy clarifications into Sec. 5.1 and Sec. B.3 of the revised manuscript.
> - We clarify that Fig. 3 computes entropy over all tokens at each diffusion step, and we update Fig. 4 to show that TSE clearly separates stable "finally correct" trajectories from oscillatory "intermediate correct" ones.
>
> We are wondering if the responses have addressed your concerns, and would appreciate it if you considered raising your final rating. Looking forward to your further feedback!

---

> ### Comment · Reviewer_CBgE · 2025-11-27
>
> Thank you for the clarification. Most of my previous concerns are resolved. However, since your TSE metric relies on syntactic extraction of the `<answer>...</answer>` span and groups intermediate outputs by exact numerical equivalence, it may be limited in capturing high-level **semantic** correctness—particularly for tasks like code generation. That said, I acknowledge the thorough experiments and the potential insights your work provides for future code-related RL research, and I support this paper's acceptance as a poster at ICLR but will maintain my score of 6.

---

> > ### Author Response · Authors · 2025-11-27
> > **Thank You for the Support and Constructive Feedback**
> >
> > Dear Reviewer **CBgE**,
> >
> > Thank you for the clarification and for supporting the paper’s acceptance. We agree that relying on syntactic extraction and exact-match grouping constrains TSE’s applicability in more open-ended generation tasks. Alternative semantic clustering approaches, such as bidirectional entailment in the spirit of Semantic Uncertainty [4] or feature-level semantic similarity, offer promising directions for extending TSE beyond exact-answer settings. We plan to explore these semantic-aware extensions in future work.
> >
> > Thank you again for the constructive feedback and for helping shape promising next steps in this research direction.
> >
> >
> >  [4] *Semantic Uncertainty: Linguistic Invariances for Uncertainty Estimation in Natural Language Generation*

---

### Author Response · Authors · 2025-11-25
**Summary of Responses to Reviewers**

We thank all reviewers for their valuable feedback and are encouraged that they found our motivation clear and the study of temporal oscillation both novel and meaningful (**CBgE**, **TojT**, **B4qA**, **39aQ**). Reviewers highlighted the strong empirical gains across benchmarks (**CBgE**, **TojT**, **39aQ**). Here, we provide a high-level summary of the changes we've made to address your concerns, and conclude with an overview of our key contributions.

---
**Here is the summary of updates that we've made to address the reviewers' concerns**:
- We expand empirical validation on new tasks. (**TojT**, **39aQ**)
  - We add experiments on two natural-language QA benchmarks (ARC-C and Winogrande).
  - We show that both Temporal Self-Consistency Voting and Temporal Consistency Reinforcement generalize beyond math reasoning tasks (Table S4, S5).
- We explore alternatives to Temporal Semantic Entropy. (**TojT**, **B4qA**, **39aQ**)
  - We evaluate other temporal-consistency metrics, including pairwise agreement (PA) and token-level answer entropy.
  - We analyze their behavior and discuss future directions for designing better temporal-dynamics metrics in dLLMs (Table S7; Fig. S5).
- We analyze the causes of temporal oscillation. (**TojT**, **39aQ**)
  - We study the effects of randomness, temperature, and different remasking strategies.
  - We show that Pass@1 vs. EverPass@1|t gap persists universally, confirming temporal oscillation as an inherent property of dLLMs (Table S2).
  - We provide a training-dynamics-based explanation in Sec. C.5.
- We provide additional analyses for Temporal Self-Consistency Voting.
  - We added a time-cost analysis showing that voting introduces only ~1% runtime overhead in Table S1. (**CBgE**, **TojT**)
  - We evaluate performance stability across multiple runs, reporting the mean and standard deviation in Table 1. (**B4qA**)
- We improve writing quality and clarity.
  - We enhance the mathematical rigor of formulas and fix minor typos. (**B4qA**, **39aQ**)
  - We strengthen the discussion of concurrent and related work. (**TojT**, **39aQ**)
  - We include more details on the decoding configuration and answer-extraction implementation. (**CBgE**, **B4qA**)

---
**The contribution of our work is summarized as follows.**
- Our work reveals a critical decoding issue in diffusion language models—**temporal oscillation**, where correct answers often emerge in intermediate denoising steps but are later overwritten. This finding challenges the common practice of relying solely on the final prediction and highlights the untapped value of temporal signals in the sampling trajectory.
- We introduce **Temporal Self-Consistency Voting**, a lightweight test-time decoding strategy that aggregates intermediate predictions to select the most temporally stable answer. This method adds virtually no inference overhead while consistently improving accuracy across multiple reasoning benchmarks.
- We further propose **Temporal Consistency Reinforcement**, a reinforcement fine-tuning approach that rewards trajectories exhibiting stable intermediate semantics. This unsupervised signal markedly boosts model performance, and can be combined with accuracy rewards to achieve even larger gains.

Together, these techniques demonstrate that **treating time as a feature** unlocks substantial improvements in reasoning stability and accuracy for diffusion language models, offering a new perspective for decoding and post-training in dLLMs.

---

### Meta-Review · Area_Chair_BewX · 2026-01-06

**Summary:**

The paper identifies a critical phenomenon in diffusion Large Language Models (dLLMs) termed "temporal oscillation," where correct answers generated during intermediate denoising steps are often overwritten in the final output. To harness these latent signals, the authors propose two complementary strategies: Temporal Self-Consistency Voting, a training-free inference method that aggregates intermediate predictions, and Temporal Consistency Reinforcement, a post-training alignment method using Temporal Semantic Entropy (TSE) as a reward signal.

Reviewers were unanimous in their appreciation of the novel perspective—treating the diffusion time steps as a feature rather than just a computation cost. The empirical results were strong, showing significant gains on math benchmarks (e.g., +25.3% on Countdown). During the rebuttal, the authors extensively addressed concerns regarding generalizability by adding Natural Language Processing benchmarks (ARC-C, Winogrande), validated the robustness of the TSE metric against alternatives like Pairwise Agreement, and rigorously clarified the mathematical formulation and answer extraction processes.

The paper offers a simple yet effective set of tools that leverage the unique properties of diffusion generation, marking a meaningful contribution to the efficient deployment and alignment of dLLMs.

**Reviewer Concerns:**

Addressed Concerns: (1) Generalizability: Reviewers 39aQ and TojT requested evaluation beyond math tasks. The authors added experiments on ARC-C and Winogrande, demonstrating consistent improvements in natural language QA.
(2) Mechanism of Oscillation: Reviewer TojT asked for a deeper explanation of why oscillation occurs. The authors provided a detailed analysis linking it to ambiguity in supervision and the "decode-before-remask" nature of the generation process, supported by controlled ablations on temperature and remasking strategies.
(3) Metric Validity: Reviewers TojT and B4qA questioned if TSE was the best metric. The authors compared TSE against Pairwise Agreement (PA) and Token Entropy, showing that TSE (and the combined TSE+Accuracy reward) yields the most robust performance.
(4) Mathematical Rigor: Reviewer B4qA pointed out notation inconsistencies. The authors thoroughly revised Equation 1 and the notation system to be probabilistically sound.

Outstanding Concerns: Applicability to Open-Ended Tasks: Reviewer CBgE noted that the exact-match clustering used for TSE might be limited in highly open-ended tasks like code generation. The authors acknowledged this as a direction for future work involving semantic entailment clustering.

**Reviewer Scores:**

Reviewer CBgE: 6

Reviewer TojT: 6

Reviewer B4qA: 4 $\rightarrow$ 6

Reviewer 39aQ: 8

---

### Decision · Program_Chairs · 2026-01-26

Accept (Poster)